# Assessing evapotranspiration dynamics across central Europe in the context of land-atmosphere drivers

Anke Fluhrer[1,2], Martin J. Baur[3], María Piles[4], Bagher Bayat[5], Mehdi Rahmati[5], David Chaparro[1,6], Clémence Dubois[7,8], Florian M. Hellwig[1,7], Carsten Montzka[5], Angelika Kübert[9], Marlin M. Mueller[7,8], Isabel Augscheller[1], Francois Jonard[10], Konstantin Schellenberg[7,11], Thomas Jagdhuber[1,2]

Microwaves and Radar Institute (HR), German Aerospace Center (DLR), Wessling, Germany
Institute of Geography, University of Augsburg, Augsburg, Germany
Department of Geography, Cambridge University, Cambridge, UK
Image Processing Laboratory, University of Valencia, Valencia, Spain
Institute of Bio-and Geosciences: Agrosphere (IBG-3), Forschungszentrum Jülich, Jülich, Germany
Centre for Ecological Research and Forestry Applications (CREAF), Cerdanyola del Vallès, Spain.
Department for Earth Observation, Friedrich Schiller University Jena, Jena, Germany
Institute of Data Science (DW), German Aerospace Center (DLR), Jena, Germany
Institute for Atmospheric and Earth System Research (INAR)/Physics, University of Helsinki, Helsinki, Finland
Earth Observation and Ecosystem Modelling Laboratory, University of Liège, Liège, Belgium
Department of Biogeochemical Processes, Max-Planck Institute for Biogeochemistry, Jena, Germany

*Correspondence to*: Anke Fluhrer (Anke.Fluhrer@dlr.de)

*Keywords*: ICOS, Eddy-covariance, MODIS, SEVIRI, ERA5-land, GLDAS-2, GLEAM, soil moisture, vapor pressure deficit, extended triple collocation, error cross-correlation, anomaly, binning

**Abstract.**

Evapotranspiration (ET) is an important variable for analysing ecosystems, biophysical processes, and drought-related changes in the soil-plant-atmosphere system. In this study, we evaluated freely available ET products from satellite remote sensing (i.e., MODIS, SEVIRI, and GLEAM) as well as modelling and reanalysis (i.e., ERA5-land and GLDAS-2) together with in-situ observations at eight Integrated Carbon Observation System (ICOS) stations across central Europe between 2017 and 2020. The land cover at the selected ICOS stations ranged from deciduous broad-leaved, evergreen needle-leaved, and mixed forests to agriculture. Trends in ET were analysed together with soil moisture (SM) from the Soil Moisture Active Passive (SMAP) mission and water vapor pressure deficit (VPD) from FLUXNET field measurements during four years including a severe summer drought in 2018, but contrasting wet conditions in 2017. The analyses revealed the increased atmospheric aridity and decreased water supply for plant transpiration under drought conditions, showing that ET was generally lower and VPD higher in 2018 compared to 2017. Across the study period, results indicate that during moisture limited drought years, ET is strongly decreasing due to decreasing SM and increasing VPD. However, during normal or rather wet years, when SM is not limited, ET is mainly controlled by VPD, and hence, the atmospheric demand.

The comparison of the different ET products based on time series, statistics, and extended triple collocation (ETC) shows in general a good agreement with ETC correlations between 0.39 and 0.99 as well as root-mean-square errors lower than 1.07 mm/day. The greatest deviations are found at the agricultural-managed sites Selhausen (Germany) and Bilos (France), with the former also showing the highest potential dependencies (error cross-correlation (ECC)) between the ET products (up to 7.6 and outside the acceptable range of -0.5 < ECC < 0.5). Hence, our results indicate that ET products differ most at stations with spatio-temporal varying land cover conditions (varying crops over growing periods and between seasons). This is because complex heterogeneity in land cover complicates the estimation of ET, while ET products agree well at evergreen needle-leaved stations with less temporal changes throughout the year and between years. The ET products from SEVIRI, ERA5-land, and GLEAM performed best when compared to ICOS observations with either lowest errors or highest correlations.

## 1 Introduction

Land-atmosphere dynamics and interactions are of key importance for understanding exchange processes in the global water, energy, and carbon cycles (Zhou et al., 2016). For a holistic and well-founded ecosystem survey, the uptake, consumption, and release of matter and energy need to be monitored. Especially in times of climate change, availability of terrestrial water, agricultural productivity assuring food security, as well as forest health guaranteeing, for instance, carbon uptake and biodiversity preservation, are mainly monitored by soil moisture (SM) and water vapor pressure deficit (VPD; as measure for atmospheric aridity) (Novick et al., 2016; Zhou et al., 2019; Liu et al., 2020). Many studies focus on these two variables when analysing drought-related terrestrial ecosystem productivity and its spatio-temporal changes (Fu et al., 2022; Zhang et al., 2021). Evapotranspiration (ET) is an important proxy for analysing water stress and its effects on ecosystems since precipitation (P) and evaporation are the two key components of the global water cycle (Miralles et al., 2011). As the sum of evaporation from land, vegetation, and water surfaces as well as transpiration from vegetation, ET directly links the terrestrial energy, water, and carbon cycles (Zhang et al., 2016; Zhou et al., 2016), and integrates meteorological conditions along SM (Bayat et al., 2022). Hence, ET is an important variable for quantifying biophysical processes, ecosystem functioning, land surface energy and water budgets, as well as improving weather and climate model predictions (Bayat et al., 2024; Zhang et al., 2016; Zhou et al., 2016). For example, Zhou et al., (2019) reported negative SM-VPD coupling, meaning low SM and high VPD, due to land-atmosphere feedbacks, since high VPD stimulates ET, which reduces SM. Although there is a debate that ET alone does not determine SM, and other factors such as precipitation should also be considered, as reduced P for constant ET can lead to lower SM (Rahmati et al., 2023), ET should in any case be one of the essential variables to inform about ecosystem-atmosphere dynamics and interactions along with SM and VPD (Bayat et al., 2021).

ET is controlled by biological (e.g., plant growth and plant stomatal regulation) and physical (e.g., temperature) processes. For example, vegetation controls interannual changes and affects spatio-temporal patterns and trends in ET (Zhang et al., 2016). ET can be theoretically linked to the independent physical control factors demand (humidity, temperature) and supply (precipitation). Depending on environmental and meteorological conditions, ET is primarily influenced by one of these three

factors. For instance, across central Europe, ET is mainly driven by the available energy due to reduced solar radiation during cloudy skies (Zhang et al., 2016). However, Seneviratne et al., (2010) stated that decreasing SM leads to decreasing ET due to the less accessible SM for plant water uptake and increasing soil suction.

During summer 2018, Europe experienced an unprecedented drought event comparable to previous extreme droughts, such as in 2003 and 2010, with a temperature anomaly of +2.8 K (Rakovec et al., 2022) and an abnormally reduced SM and increased VPD (Fu et al., 2022). This extreme drought was characterized by a unique geographical distribution, focused on regions at higher latitudes (central and northern Europe), a rapid change from a wet spring to a dry summer, and an intense heatwave in the spring of 2018 (Bastos et al., 2020). As a result, it caused severe tree stress in central Europe, with low leaf water potential, leaf discolouration, and premature shedding, leading to significant tree mortality and heavy drought-legacy effects in 2019, leaving trees vulnerable to further damage from pests and pathogens (Schuldt et al., 2020).

The significance of ET can also be seen in relation to the precise parametrization of SM and its memory in Land Surface Models (LSMs) (Rahmati et al., 2024). Due to its importance and influence on the entire soil-plant-atmosphere system (SPAS), tracking ET in time and space, meaning from seasonal to multi-year scales and for wide areas, is necessary and calls for a satellite remote sensing approach (complementary to current modelling and reanalysis approaches). Although it is not directly measurable from remote sensing acquisitions, optical, thermal, infrared, or microwave observations are used to derive ET based on surface energy balance, physical and empirical models (Zhang et al., 2016; Rahmati et al., 2020; Singh et al., 2020; Bayat et al., 2021; Bhattacharya et al., 2022; Bayat et al., 2024). Still, research comparing the performance of remote sensing with model and reanalysis data under drought conditions is lacking, and an analysis on how main ET drivers (SM and VPD) impact these ET products is also needed. Bridging this gap is paramount to assess which products and in which conditions are more suitable to track ET, especially under the increasingly frequency and severity of droughts due to climate change.

Several regional studies exist for comparing various ET products, e.g., over China (Meng et al., 2024; Xu et al., 2024), across the U.S. (Carter et al., 2018; Xu et al., 2019), over Africa (Trambauer et al., 2014), and across Europe (Ahmed et al., 2020; Stisen et al., 2021). However, due to the complexity of ecosystems, findings from specific regions (e.g., China, U.S., Africa) cannot be generalized for other regions (e.g., Europe). Further, European studies focused either only on spatial product comparisons, evaluating the performance of hydrological models (e.g., Stisen et al., 2021), on former time periods (e.g., 2003-2013) at basin scale (Liu et al., 2023), on analysing drought impacts on ET dynamics using solely a single ET product (e.g., Sepulcre-Canto et al., 2014; Ahmed et al., 2020), and on evaluating new ET products (e.g., Hu et al., 2023). For example, the focus in the study of Stisen et al., was the evaluation of the spatial pattern performance in different hydrological models for ET estimation. For this, four remote sensing based ET products were inter-compared for years 2002-2014, and they found high agreements in spatial patterns across continental Europe (Stisen et al., 2021). Further, Ahmed et al., investigated the drought impact of 2018 on the MODerate Resolution Imaging Spectroradiometer (MODIS) ET across European ecosystems and found that ET decreased up to 50% compared to a 10-year reference period, with agricultural areas and mixed natural vegetation being most affected (Ahmed et al., 2020). However, there is a lack of studies comparing various ET products among each other and with in-situ measurements across central Europe, especially during severe drought years (e.g., 2018), as well as

evaluating the potential of remote sensing for tracking seasonal ET dynamics. The evaluation of the varying employed retrieval
techniques (e.g., eddy covariance, land surface schemes, Penman-Monteith equation) of commonly used ET products under
drought conditions is paramount in order to assess whether they capture ET dynamics correctly.
In this study, we first compare the most common ET products from field measurements, modelling, and remote sensing across
central Europe for the period 2017 to 2020. These selected six products (ICOS, MODIS, SEVIRI, ERA5-land, GLDAS-2, and
GLEAM) are well-known, commonly employed, and freely available. The focus hereby is on the evaluation and quality
assessment of the individual products regarding the estimation of absolute ET values and their time-dynamics. Second, we
compare ET products in the context of SM and VPD, disentangling the relative role of all three variables within the SPAS
under severe drought conditions in 2018 in comparison to the rather wet year 2017. This is to analyse how the ET products
catch drought conditions and to what extent they can be used as indicator for drought events.
**2 Materials and Methods**
**2.1 Study area**
The focus is on eight Integrated Carbon Observation System (ICOS) (Rebmann et al., 2018) stations within central Europe
between 2017 and 2020, where field-scale in-situ eddy-covariance (EC) ET measurements are available (see Fig. 1).

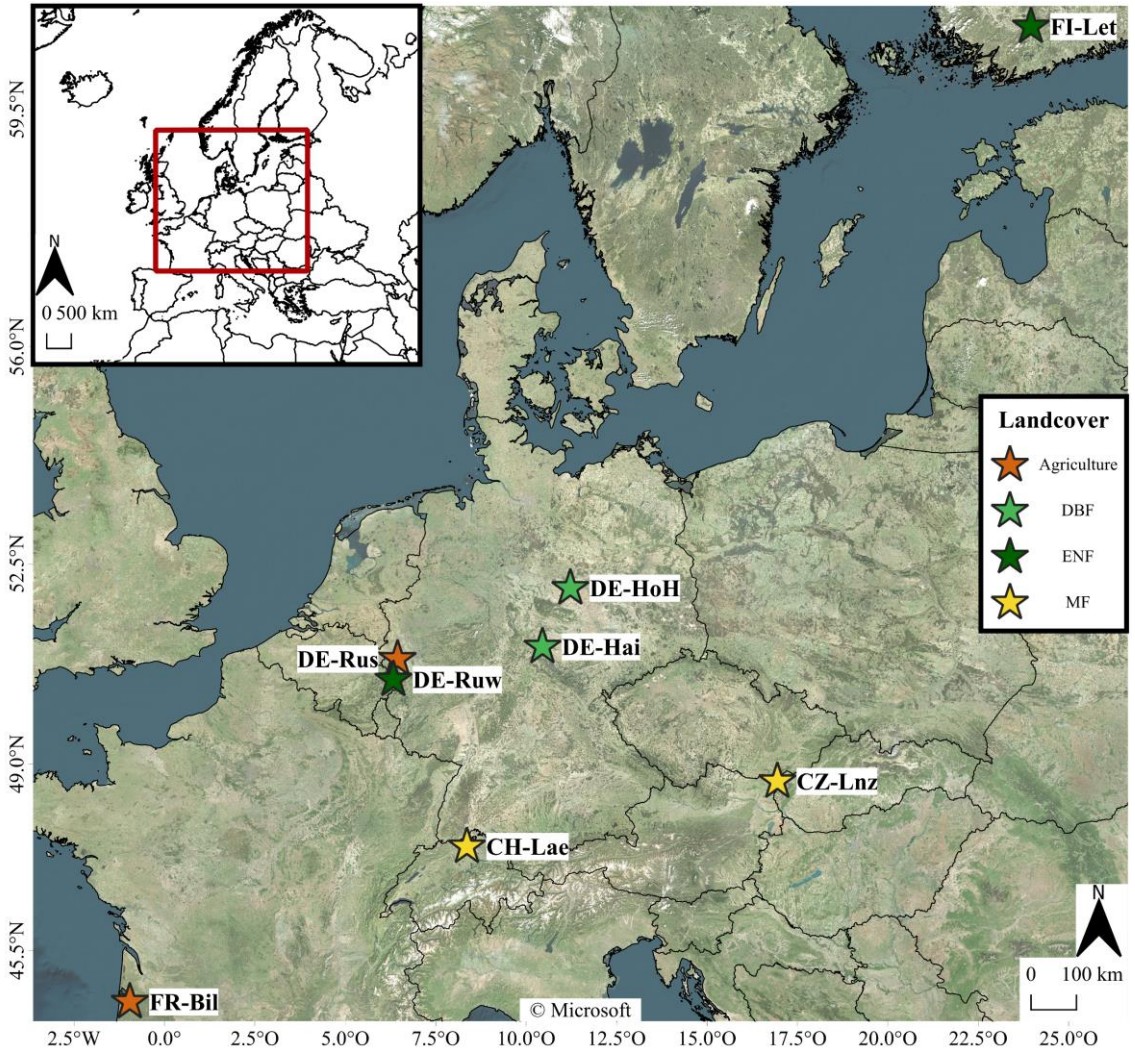


Figure 1: Location of the eight investigated Integrated Carbon Observation System (ICOS) stations in central Europe, and their classification
according to the respective dominant land cover class. DBF = deciduous broad-leaved, ENF = evergreen needle-leaved, MF = mixed forest.
The study comprises two deciduous broad-leaved (DBF) — the German Hohes Holz (DE-HoH) and Hainich (DE-Hai), two
evergreen needle-leaved (ENF) — the German Wuestebach (DE-Ruw) and Finnish Lettosuo (FI-Let), and two mixed forest
(MF) stations — the Czech Landzhot (CZ-Lnz) and the Swiss Laegern (CH-Lae), as well as two agriculture stations — the
German Selhausen (DE-Rus) and the French Bilos (FR-Bil). Details regarding coordinates, altitude, and climate zone for every
station are given in Table S1. At every station, a footprint of 3 km radius is analysed to account for differences in spatial
resolutions among employed datasets (see Sec. 2.2). As displayed in Figure 2 and Table S2 (supplement), the land cover types
and their homogeneity within the 3 km × 3 km footprint around every station was analysed based on the Corine land cover
(CLC) 2018 classification from the Copernicus Land Monitoring Service at 100 m spatial resolution (European Environment
Agency, 2019).

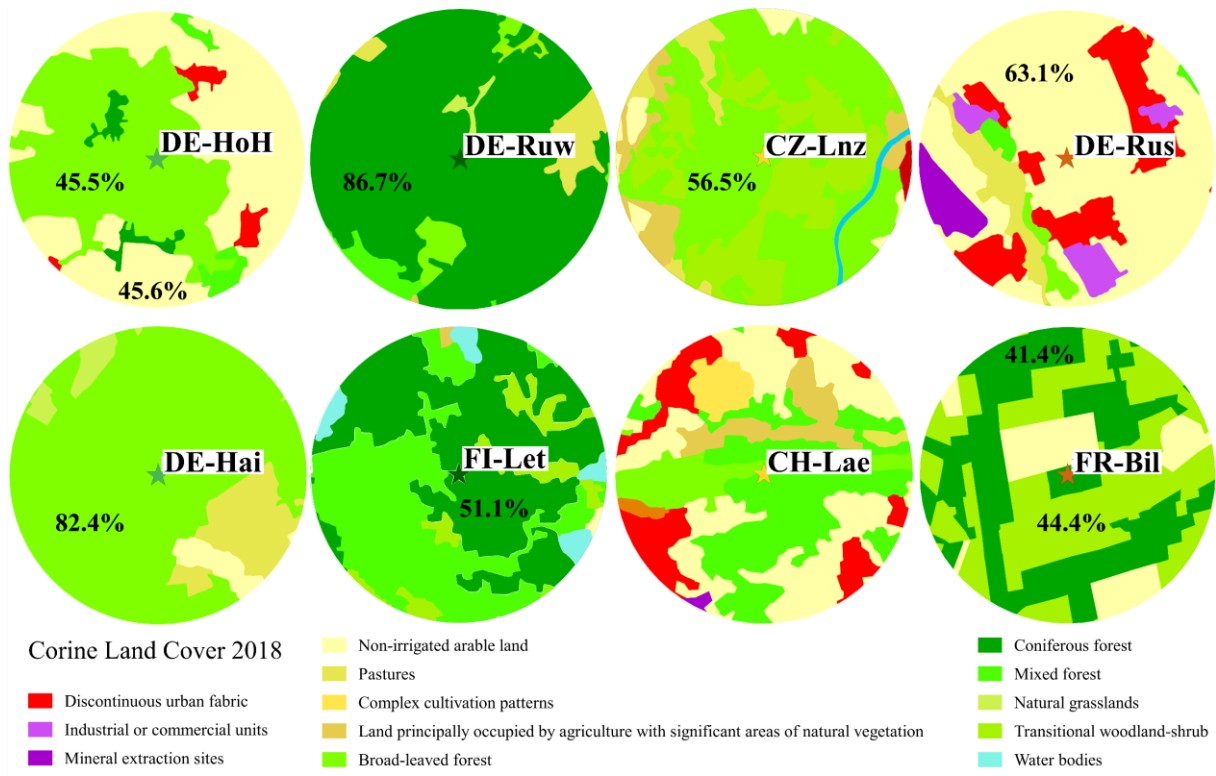

Figure 2: Overview of land cover classes according to the Corine Land Cover (CLC) 2018 (European Environment Agency, 2019) within the 3 km × 3 km footprint around every investigated ICOS station. Percentages inside the circles indicate the dominant land cover classes, respectively. The percentages of all land cover classes at every station can be found in the supplement (see Tab. S2).

According to this classification, two stations can be considered as homogeneous with one dominant land cover class, i.e., 86.7 % of coniferous forest at DE-Ruw, and 82.4 % of broad-leaved forest at DE-Hai. Station DE-Rus is mainly (63.1 %) covered by non-irrigated arable land. Further, two stations show a two-part split land cover with two almost equally dominant classes. At DE-HoH, 45.6 % are covered by non-irrigated arable land and 45.5 % are covered by broad-leaved forest. At FR-Bil, although it is officially labelled as ENF station, 44.4 % are covered by transitional woodland shrub, while 41.4 % are covered by coniferous forest, a managed Pine forest plantation (Loustau et al., 2022). Hence, due to this heterogeneity and the fact that 14.2 % of non-irrigated arable land (see Tab. S2) are mostly directly located near the station (see Fig. 2), we ranked it as agricultural station in order to account for the frequently changing land cover conditions and spatial heterogeneity. All other stations are rather heterogeneous with a mix of more than two different land cover classes (see Tab. S2 and Fig. 2). However, it is worth noting that the CLC 2018 classification is based on data from 2017 to 2018. Hence, changes in the land cover, e.g., such as differences between summer and winter months, deforestation, weather extremes (storms, floods), or varying agricultural crop cultivation, at each station between 2017 to 2020 are not included here.

Figure 3 illustrates the meteorological conditions (precipitation P and air temperature $T_{Air}$) at every station during the
investigation period. The mean annual P and $T_{Air}$ values are summarized in Table S1. Note that the in-situ P measurements
contain missing values at stations DE-HoH, CZ-Lnz, and CH-Lae in 2020. The overall lowest $T_{Air}$ is found at the northernmost
ICOS station FI-Let, varying between -12.6 °C (absolute minimum) and 22.75 °C (absolute maximum) in the years 2017 to
2020, with an interannual average of 5.67 °C. In contrast, the highest average $T_{Air}$ (between 2017 and 2020) of 14.1 °C is found
at the southernmost ICOS station FR-Bil, which also has the highest average P value of 3.04 mm/day. The lowest P is found
at DE-HoH with an average of 1.26 mm/day, which is similar to the other stations in the mid-latitudes (see Tab. S1). The
overall highest $T_{Air}$ and lowest P at every station are always found in 2018 with an average of 1.7°C higher $T_{Air}$ and annual
0.76 mm higher P, compared to the second hottest and driest year in each case. Exceptions can be found at the station FR-Bil,
where the highest $T_{Air}$ are recorded in 2019 and lowest P in 2017, and DE-Ruw, as well as CH-Lae, where the lowest average
annual P are recorded in 2020, respectively.
Based on the standardized precipitation-evapotranspiration index (SPEI) (Beguería et al., 2023) (see Fig. S1), which describes
drought based on the amount and duration of water deficit (Yu et al., 2023), distinctly dry and wet years are identified for each
ICOS station. While all stations show abnormally dry periods, especially for 2018, only stations FI-Let and FR-Bil show
abnormally wet periods at the end of 2017 and 2019. These two are the northernmost and southernmost stations (see Fig. 1).
The choice of SPEI to identify drought conditions instead of the standardized precipitation index (SPI) or other indices (i.e.,
Palmer drought severity index) is due to the fact that the SPEI considers implicitly temporal changes in ET and hence,
temperature, which is relevant for identifying abnormal (drought) conditions and for this study with focus on ET variations.
Previous studies showed that not only the lack of precipitation defines drought events but also the level of temperature and
consumption of rainfall by evaporation and/or transpiration (Vicente-Serrano et al., 2010).

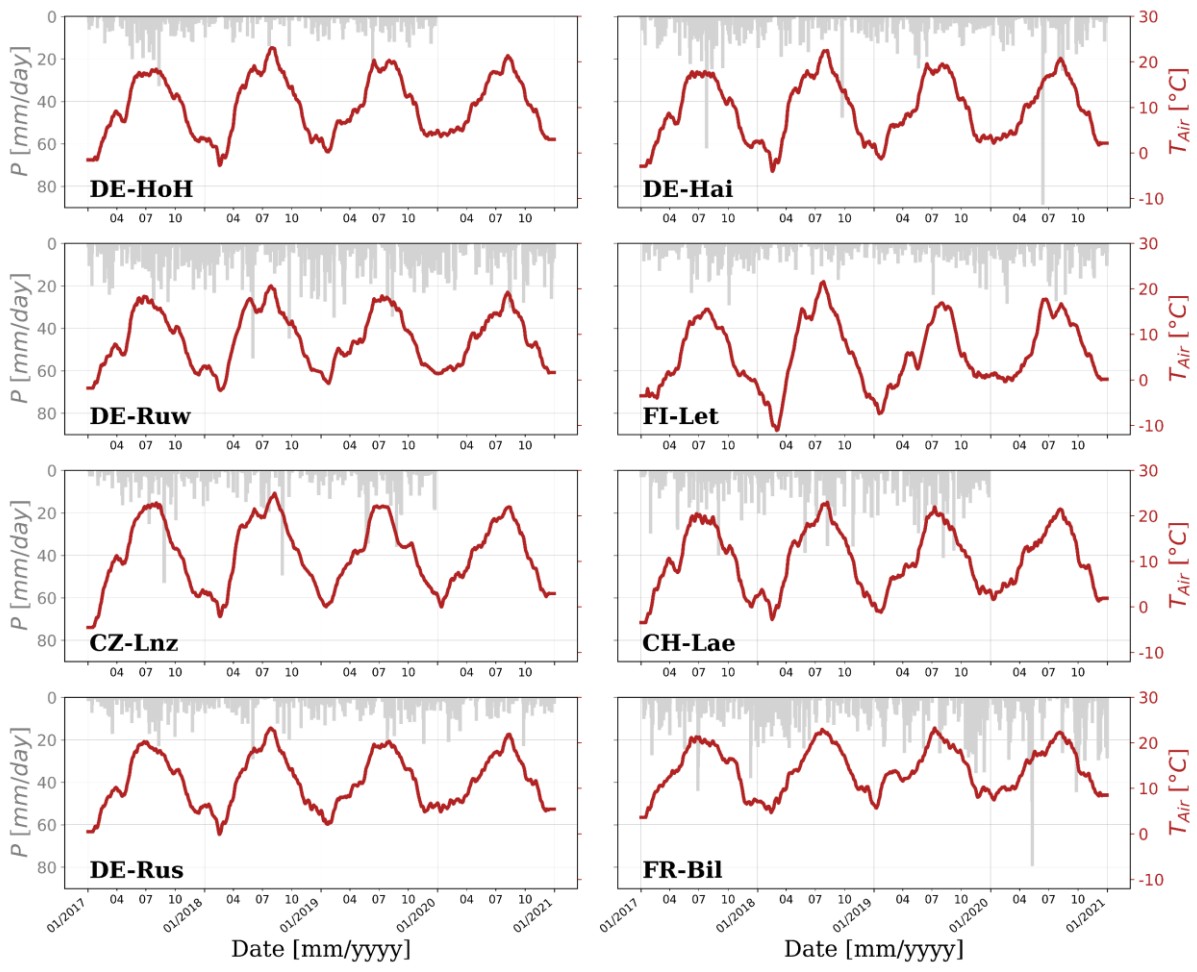


Figure 3: Daily in-situ measured precipitation (P) [mm/day] and air temperature ($T_{Air}$) [°C] at investigated ICOS stations. $T_{Air}$ was cleaned for daily and weekly dynamics using a Savitzky-Golay (Savitzky and Golay, 1964) filter with a window size of 31 days.

**2.2 Data base**

In the first part of this study, different ET products (see Tab. 1) are inter-compared in order to evaluate the potential of remote sensing for tracking seasonal ET dynamics. The in-situ ET data, recorded at the ICOS stations at field-scale, are mass balance-based measurements of sensible heat (H) and latent heat (LE) fluxes through the covariance of heat and moisture fluxes, respectively. The LE [W/m$^2$] can then be converted to ET by dividing it by the latent heat of vaporization (2.434 [MJ/kg] at 20 °C air temperature) (Allen et al., 1998). The ICOS network has undertaken significant efforts to ensure consistent high-quality LE measurements across stations (Rebmann et al., 2018). Besides in-situ ET measurements, we employ some of the most commonly employed optical/thermal remote sensing products from NASA's (National Aeronautics and Space Administration) Moderate-resolution Imaging Spectroradiometer (MODIS) sensor on Terra (Running et al., 2017), ESA's Spinning Enhanced Visible and Infrared Imager (SEVIRI) sensor onboard of the Meteosat Second Generation (MSG) satellites,

and the Global Land Evaporation Amsterdam Model (GLEAM) (Martens et al., 2017). Further, also well-known reanalysis and modelling products from the land component of the Earth system modelling product European Re-Analysis (ERA5-land) from the European Centre for Medium-Range Weather Forecasts (ECMWF) (Muñoz Sabater, 2019), and from NASA's Global Land Data Assimilation System Version 2 (GLDAS-2) (Beaudoing et al., 2020) are used (see Tab. 1). It should be noted that the GLEAM product is based on various remote sensing observations and reanalysis datasets from, e.g., NASA's SMOS (soil moisture and ocean salinity) mission, MODIS, GLDAS-Noah, and ERA-Interim (Martens et al., 2017). The MODIS product with nominal spatial resolution of 500 m is aggregated to the 3 km footprint, while the SEVIRI, ERA5-land, GLDAS-2, and GLEAM products are maintained at their original spatial resolutions of 3 km, 9 km and 25 km, respectively. Although several downscaling methods and data fusion techniques exist for improving the spatial resolution of remote sensing products (Ha et al., 2013; Mahour et al., 2017; Peng et al., 2017), we decided to keep ET products with a spatial resolution lower than 3 km at their original resolution (i.e., GLEAM at 25 km). For one, the intention of this study is a comparison of well-known and established ET products and not an optimization of rescaled comparisons. Second, we did not want to include additional uncertainties potentially originating from the employed downscaling method or auxiliary datasets. Especially, downscaling approaches intend to statistically correlate coarse-scale data and fine-scale auxiliaries, yielding to interpolation uncertainties and errors that cannot be quantified (Peng et al., 2017). All datasets are, however, temporally aggregated to daily time series in order to provide a temporal basis for comparison and analysis of the signal dynamics.

Table 1: Overview of investigated ET and auxiliary products presenting the data source, the original spatial and temporal resolution as well as the retrieval basis and method of each product.

| PRODUCT (NAME) | SOURCE | ORIGINAL SPATIAL / TEMPORAL RESOLUTION | RETRIEVAL BASIS | RETRIEVAL METHOD |
|---|---|---|---|---|
| ET PRODUCTS | | | | |
| ICOS (Level 2) | ICOS (ICOS RI et al., 2024) | Point scale / Half-hourly | In-situ measurements | Eddy covariance technique |
| MODIS (MOD16A2) | NASA (Running et al., 2017) | 500m / 8-daily | Remote Sensing | Penman-Monteith |
| ERA5-land | ECMWF (Muñoz Sabater, 2019) | 9 km / hourly | Reanalysis | ECMWF's IFS, H-TESSEL land surface scheme |
| SEVIRI (METv3) | ESA (LSA SAF and EUMETSAT SAF On Land Surface Analysis, 2019) | 3 km / half-hourly | Remote Sensing | SVAT, (H-) TESSEL land surface scheme |

| | | | | |
|---|---|---|---|---|
| **GLDAS-2 (GLDAS_NOAH 025_3H_2.1)** | NASA (Beaudoing et al., 2020) | 25 km / 3-hourly | Land Surface Model (NOAH) L4 | Penman-Monteith |
| **GLEAM (v3)** | University of Amsterdam (Miralles et al., 2011; Martens et al., 2017) | 25 km / daily | Remote Sensing | Priestley-Taylor |
| **AUXILIARY PRODUCTS** | | | | |
| **FLUXNET2015** | (Pastorello et al., 2020; Warm Winter 2020 Team et al., 2022) | Point scale / half-hourly | In-situ measurements / Reanalysis | Downscaled and consolidated from ERA5-interim reanalysis data and gap filled |
| **SMAP MT-DCA V5** | (Feldman et al., 2021) | 9 km / daily | Remote Sensing | Tau-Omega; Multi-temporal dual channel algorithm (MT-DCA) |
| **SPEI V2.8** | (Beguería et al., 2023) | 0.5° / 3-monthly | Remote Sensing / Modelling | FAO-56 Penman-Monteith method |

In Table 1, the retrieval methods for each ET product are given. MODIS and GLDAS-2 are based on physically-based methods
employing the Penman-Monteith equation (Penman, 1948; Monteith, 1965), while GLEAM is based on the Priestley-Taylor
equation (Priestley and Taylor, 1972), and ERA5-land uses the ECMWF integrated forecasting system (IFS) and is derived
from the ERA5 product where the land surface model is based on the hydrology Tiled ECMWF Surface Scheme for Exchange
Processes over Land (H-TESSEL) (Hersbach et al., 2020). Further, SEVIRI employs a soil-vegetation-atmosphere-transfer
(SVAT) approach also based on the physics of the TESSEL and H-TESSEL land surface scheme (Balsamo et al., 2009; Bayat
et al., 2024; Ghilain et al., 2011). The officially reported ET biases after evaluation of each product (based on comparison with
multiple EC flux tower measurements) range from -0.11 mm/day (MODIS) (Running et al., 2019) and -0.12 mm/day (SEVIRI)
(The EUMETSAT Satellite Application Facility on Land Surface Analysis (LSA SAF), 2024), to -5% (GLEAM) (Miralles et
al., 2011). Meaning, all three products show in average slightly lower ET values compared to EC flux tower measurements.
All other products indicate no bias, but employ either bias corrected atmospheric reanalysis data for the forcing to avoid
discontinuity in ET retrievals (GLDAS-2) (Rui and Beaudoing, 2022), or found no significant difference in comparison to
other products (ERA-land) (Muñoz-Sabater et al., 2021). The Priestley-Taylor equation does not consider the impact of VPD
or canopy conductance (Wang and Dickinson, 2012), while within the Penman-Monteith equation VPD and relative humidity
(RH) are used according to the function of Fisher et al., (2008) in order to account for soil water stress when calculating the
actual soil evaporation. Further, the canopy conductance is retrieved from stomatal and cuticular conductance depending on

LAI and the wet surface fraction, with the stomatal conductance constrained by VPD and minimum air temperature and the cuticular conductance fixed to a constant of 0.01 [mm/s] (Running et al., 2019; Wang and Dickinson, 2012). As stated by (He et al., 2022), the Penman-Monteith equation includes the most important modification by accounting for the physiological controls on ET, using stomatal resistance to explain water movement from leaves to the atmosphere and aerodynamic resistance to describe heat and water vapor transfer from the dry canopy surface to the air above (Running et al., 2019). Hence, the Penman-Monteith equation is, in theory, more accurate than the Priestley-Taylor equation but, in turn, requires more 'parameters that are difficult to characterize' (Fisher et al., 2008). Within the TESSEL and H-TESSEL schemes, canopy conductance is formulated according to the modified Jarvis function and based on the stomatal conductance (retrieved from net assimilation and Kirchhoff's resistance/conductance analogy) and cuticular conductance (fixed between 0 to 0.25 [mm/s] according to vegetation types), while SM at four layers, and therefore also deeper soil layers, are accounted when defining the soil water stress on soil evaporation (ECMWF, 2018). Lastly, for this study, it is interesting to note that GLEAM and ERA5-land employ the ECMWF atmospheric reanalysis data (Li et al., 2022), while GLDAS-2 is based on MODIS land surface parameters (Rui and Beaudoing, 2022). These product interdependencies should be kept in mind during interpretation of results.

In the second part of this study, the ET products are compared in relation to two dominant parameters of the SPAS, namely SM and VPD. While VPD comes from in-situ measurements of the Fluxnet network (point precise), SM comes from NASA's Soil Moisture Active Passive (SMAP) mission, the multi-temporal dual channel algorithm (MT-DCA) L-band (1.4 GHz) dataset (9 km spatial resolution) (Konings et al., 2016; Feldman et al., 2021) (see Tab. 1). We employed the SMAP SM in this study instead of using available in-situ measurements of the Fluxnet network, since the latter were of poor quality at several stations and years, and we wanted to build our analyses on one continuous dataset. The SMAP MT-DCA dataset is quality controlled and filtered for, e.g., snow, frozen ground, and water bodies (Feldman et al., 2021).

## 2.3 Methods

### 2.3.1 Extended triple collocation

For the comparison of different ET products in sec. 3.1., the extended triple collocation (ETC) method (McColl et al., 2014) is employed. The ETC technique not only provides the root-mean-square-error $\sigma_\varepsilon$ [mm/day] of the classical triple collocation (TC) method (Stoffelen, 1998) among three independent measurement systems, but also provides the correlation $\rho_{t,X}$ [-] among them, giving the sensitivity of the measuring systems. The most important advantage of the TC and ETC techniques is that one can calculate $\sigma_\varepsilon$ and $\rho_{t,X}$ without considering any of the systems as the necessary reference. The product with the lowest $\sigma_\varepsilon$ and highest $\rho_{t,X}$ identifies the one with the lowest uncertainty. As input to the ETC, the daily ET time series are filtered for the growing season (April to October) of each year. With the aim of evaluating the performance of the remote sensing products (SEVIRI, MODIS, GLEAM), we compare them individually with ERA5-land and in-situ measurements (ICOS) on the one hand, and with GLDAS-2 and ICOS on the other hand. Sanity checks for Gaussian distributions and large sample sizes of

~853 values per product ensure precise and representative ETC analyses. Additionally, since one of the requirements for thorough ETC analyses is the independence among evaluated datasets (McColl et al., 2014), the error cross-correlation (ECC) values (Gruber et al., 2016) are calculated in order to evaluate product dependencies. In case the ECC lies between -0.5 and 0.5, the datasets can be regarded as independent from each other. The ECC for each product comparison (with ET product $\in$ [i,j,k,l]) is calculated from the error cross covariance $\sigma_{\varepsilon_i \varepsilon_j}$ between two products as well as the random error variance $\sigma_{\varepsilon_i}^2$ of each dataset, respectively (Gruber et al., 2016):

$$ECC_{ij} = \frac{\sigma_{\varepsilon_i \varepsilon_j}}{\sigma_{\varepsilon_i}^2 \sigma_{\varepsilon_j}^2}, \tag{1}$$

with

$$\sigma_{\varepsilon_i \varepsilon_j} = \sigma_{ij} - \frac{\sigma_{ik}\sigma_{jl}}{\sigma_{kl}}, \tag{2}$$

and

$$\sigma_{\varepsilon_i}^2 = \sigma_i^2 - \frac{\sigma_{ij}\sigma_{ik}}{\sigma_{jk}}. \tag{3}$$

**2.3.2 Anomalies**

For the comparison of different SPAS parameters in sec. 3.2., the seasonal imprint is removed from the signals in order to focus on exceptional events in the time series. For that, we calculated the 30-day anomaly time series for each parameter. To do so, the daily average over all four years was calculated first. The resulting daily average was then smoothed using a Savitzky-Golay (Savitzky and Golay, 1964) filter with a window size of 61 days. Lastly, for every day between 2017 to 2020, the difference between the day of interest and the 30-day average of the filtered daily average before that day has been calculated.

**2.3.3 Binning**

To analyse the effects of water supply and demand on ET, we binned daily ET values into a grid of 30 by 30 SM and VPD conditions, with SM ranging between 0.0001 vol.% and 40 vol.%, and VPD ranging between 0.0001 hPa and 25 hPa, both in 31 steps (to create a grid of 30 by 30). While SM is indicative of the available water supply, VPD is an indicator of atmospheric water demand. The co-regulation of ET by SM and VPD is complex as it depends on stomatal and surface conductance, which in turn are dependent on SM and VPD, as well as vegetation and soil properties (Carminati and Javaux, 2020; Zhang et al., 2021; Vargas Zeppetello et al., 2023). To understand the main directionality of ET changes relative to SM, we calculated the average slopes of ET relative to SM (equivalent to $\frac{\Delta ET}{\Delta SM}$). The same applies when we examine the directionality of the ET changes with respect to VPD ($\frac{\Delta ET}{\Delta VPD}$). These analyses are done in order to get an indication of the dominating control on ET.

## 3 Results

### 3.1 Differences in examined ET products

In Figure 4, times series of the employed ET products (see Tab. 1) are shown at all investigated ICOS stations (see Fig. 1) for the period 2017 to 2020. Apart from the seasonal dynamics of ET, with highest values in the summer months (June, July, August) and low values but with more frequent changes in the winter months (November, December, January), the overall good consistency between the different ET products can be noted.

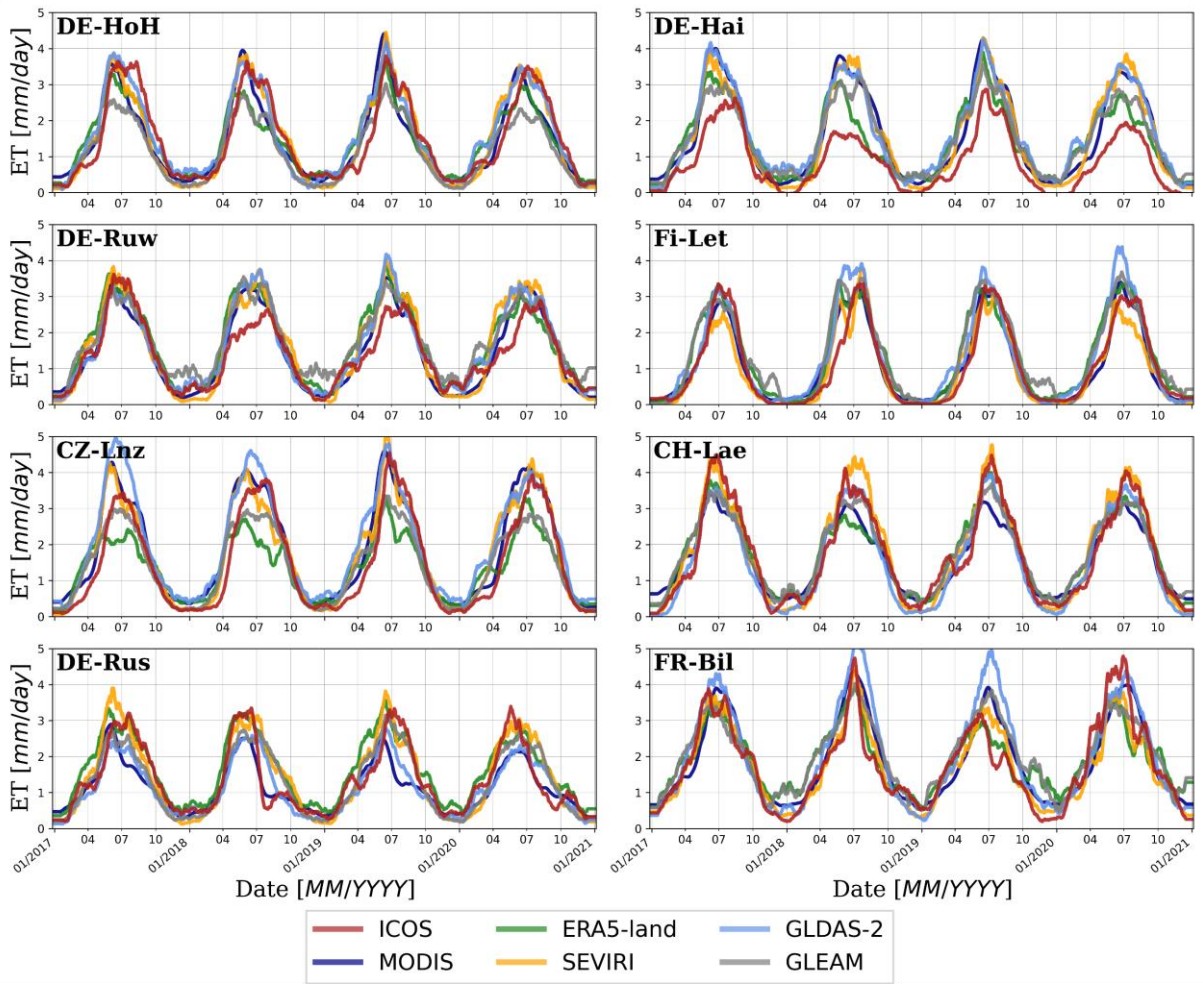

Figure 4: Comparison of seasonal dynamics of ET [mm] products for the period 2017-2020 at investigated ICOS stations. All time series were cleaned for daily and weekly dynamics using a Savitzky-Golay (Savitzky and Golay, 1964) filter with a window size of 31 days.

The highest variability among products and ET dynamics can be observed during summer months, with greatest differences at stations DE-Hai and DE-Ruw when comparing all products to the ICOS measurements. Here, the ground-based ET shows always lower values across all years for DE-Hai, and in 2018 and 2019 for DE-Ruw. Additionally, for each year, the ICOS ET

rises a few weeks later than the other products at both stations but decreases together with all other ET products. These differences and delayed seasonal increase of remote sensing, modelling, and reanalysis products compared to the ICOS measurements at the DBF and ENF station occur, for one, due to the discrepancies in spatial resolutions (point-scale versus kilometer scale). Second, ICOS field measurements provide a different sensitivity to vegetation phenology than the other remote sensing & modelling products due to measuring directly above the canopy. At station CZ-Lnz, ERA5-land shows the overall lowest ET values during the growing period (April to October). Further, the highest ET values are found at station FR-Bil for the GLDAS-2 product with most pronounced differences to all other products in 2018, while overall lowest values across all years and ET products are displayed at DE-Rus. At the latter, ET values never exceed 4 mm/day. From this daily time series analyses, the largest differences among ET products can be seen at the DBF station DE-Hai, MF station CZ-Lnz, and agriculture station DE-Rus. At DE-Hai, the ICOS ET is overestimated by all other products, at CZ-Lnz, the ERA5-land product is lower compared to all other ET products, especially in the summer months, and at DE-Rus, the MODIS and often also the ICOS product are overestimated by the ERA5-land and SEVIRI products. Hence, no clear pattern at all stations and between different land cover classes can be found.

For more detailed analyses, daily time series of ET products are averaged to 8-daily sums in order to account for the coarse temporal resolution of the MODIS product (see Tab. 1). In Figure 5, the 8-daily ET products are compared with each other at the two agriculture stations. The same illustrations for the forest stations can be found in the supplement (see Figs. S2-S4). These figures show the scatter plots between ET products giving the probability density function (PDF) of points (by colour) below (left panels) and above (right panels) the matrix diagonal, as well as the PDF curves for each site and product in the diagonal of the matrix. They support the previously stated good consistency between ET products but outline the exact differences on 8-days scale in more detail. The highest density of values can be observed between 0 to 10 mm/8-days at all stations except at DE-Ruw and FR-Bil. This comes from the rather low ET values during the autumn, winter, and spring seasons due to the overall reduced solar radiation combined with decreased vegetation cover during cold months. However, at stations DE-Ruw (see Fig. S3, right panels) and FR-Bil (see Fig. 4, left panels), the density of values is shifted towards higher ET (0 to 20 mm/8-days). These are two out of the three stations covered by coniferous forest. While FR-Bil has a two-part split land cover in the footprint (shrub and coniferous forest), DE-Ruw is almost homogeneously covered by coniferous forest (see Fig. 2), and both stations show higher ET values during autumn and spring seasons compared to all other stations due to, e.g., the lack of leaf off conditions during that periods. The third station covered by coniferous forest (FI-Let), however, shows the density of values between 0 to 10 mm/8-days (see Fig. S3, left panels), similar to DBF and MF stations. This is the northernmost station, typically covered with snow between November and March.

Further, the over- or underestimation of values between two products can be seen, such as the overestimation of ICOS compared to all other ET products at DE-Hai for higher ET values, affirmed by the PDF for ICOS peaking at the highest density (see Fig. S2, left panels). There is also an overestimation of MODIS compared to all other products at DE-Rus (see Fig. 5, right panels) and CH-Lae (see Fig. S4, left panels) when ET values are higher. DE-Rus is the only homogeneously covered agricultural station with potentially most changes in land cover classes during the seasons and years, showing the

greatest differences in ET products due to the overall higher complexity of agricultural plants and more frequent alterations.
While the PDF of MODIS at DE-Rus peaks at the highest density and gives the smallest range of ET values across all stations,
a bimodal distribution of densities is displayed at CH-Lae. This bimodal distribution of densities is also noticeable at other
products and stations but stronger always for MODIS.

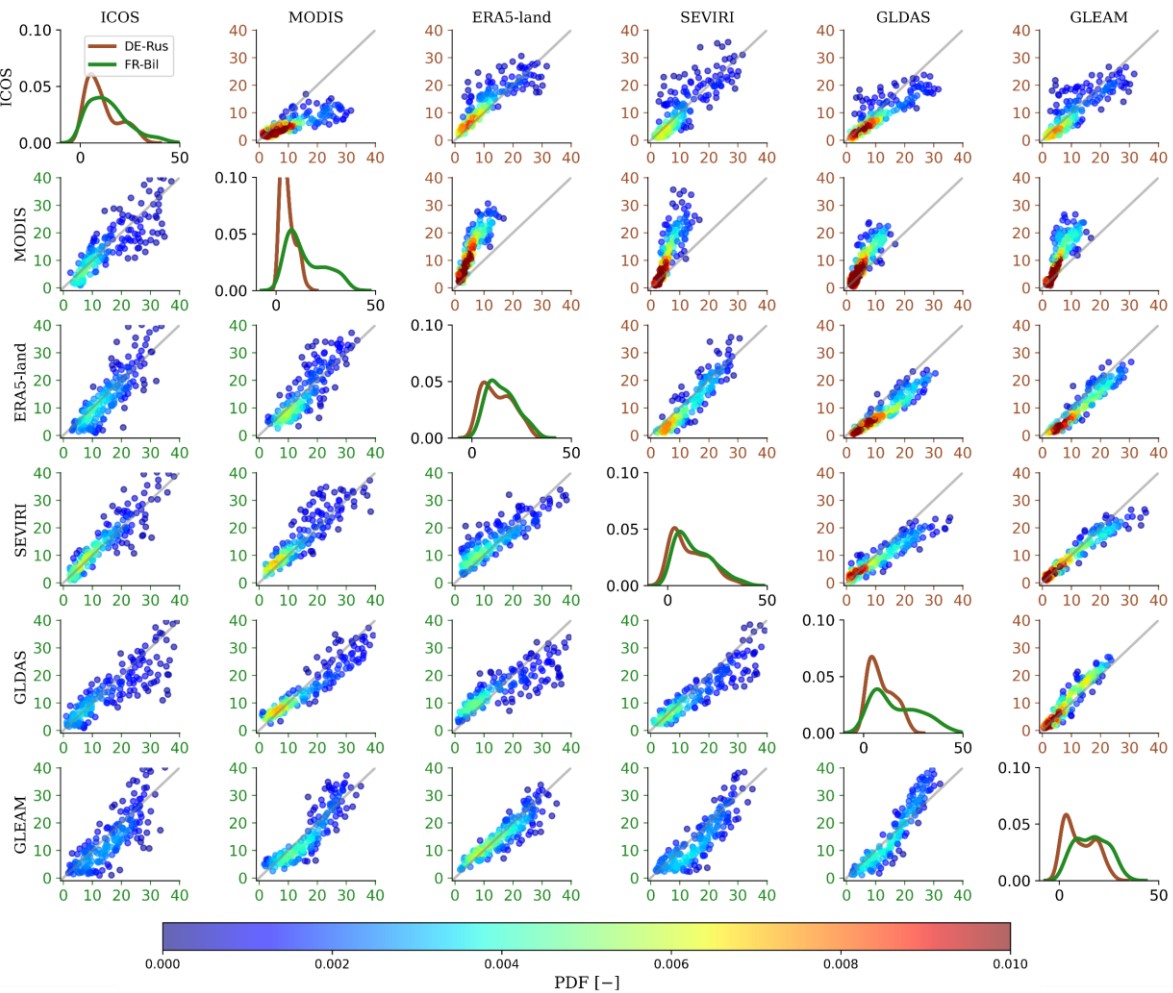


Figure 5: Comparison of seasonal dynamics of ET [mm/8-days] products for the period 2017-2020 at investigated ICOS stations DE-Rus
(right panels above the diagonal of the matrix) and FR-Bil (left panels below the diagonal of the matrix). All time series were averaged to 8-
daily sums at MODIS dates, and cleaned for daily and weekly dynamics using a Savitzky-Golay (Savitzky and Golay, 1964) filter with a
window size of 31 days. All statistics are included in supplement Figures S5-S7.
This visual interpretation is also supported by statistics in supplement Figures S5-S7. In general, the highest coefficient of
determination, $R^2$ [-], among all products can be found at station CH-Lae, while the overall lowest root-mean square errors,
RMSE [mm/8-days], are retrieved at both ENF stations (DE-Ruw, FI-Let). DE-Ruw is also the station with, in general, lowest
percentage bias, PBIAS [%], among all ET products. In detail, the highest $R^2$ of 0.94 is found between GLEAM and GLDAS-
2 at CH-Lae, while the lowest RMSE of 2.3 mm/8-days and the lowest PBIAS of -0.05 % is found between GLEAM and
ERA5-land again at CH-Lae. The lowest $R^2$ of 0.62 and highest PBIAS of 91 % is found between ICOS and MODIS at the
agricultural station DE-Rus, while the highest RMSE of 8.8 mm/8-days is found between MODIS and ERA5-land again at
DE-Rus. In summary, the statistics indicate an overall worse consistency among products at the rather mixed agricultural
station (DE-Rus) and better consistency at ENF stations.
In order to evaluate the performance of each ET product in more detail, the ETC method (McColl et al., 2014) is employed.
Here, we use the ETC approach to compare the three remote sensing products individually first with ERA5-land and ICOS,
and then with GLDAS-2 and ICOS. The preceding calculation of ECC values among all products (see Fig. S8) is conducted
to ensure the independence of the examined products, which is required by ETC analysis (see Sec. 2.3.1). Overall, ECC values
are always around zero or within the acceptable range of -0.5 to 0.5. Only at station DE-HoH between GLDAS-2 and GLEAM,
at CZ-Lnz between ERA5-land and GLEAM, at CH-Lae between ERA5-land and MODIS as well as for all product
comparisons at DE-Rus (except between ERA5-land and SEVIRI), ECC values outside the acceptable range can be found (see
Fig. S8). The high ECC values at DE-HoH, CZ-Lnz, and DE-Rus between GLEAM and GLDAS-2 or ERA5-land is not
surprising, since the GLEAM product is based on various remote sensing and reanalysis datasets, with among others GLDAS
and ERA5 (see Sec. 2.2). Hence, at most stations ET products can be regarded as statistically independent from each other.
Only some potential product dependencies, especially at the agricultural station DE-Rus, should be kept in mind during the
interpretation of ETC results.
In Figure 6, the ETC statistics for the applied product combinations at all stations are shown. While the x- and y- axes represent
the estimated root-mean-square-error $\sigma_\varepsilon$, the arcs give the correlation $\rho_{t,X}$. Hence, numbers (representing the eight stations)
close to zero on the x- and y-axes and close to one on the arcs give the best ETC results, meaning lowest uncertainty of the ET
product (represented by colours) compared to the other two products, respectively. It can be seen that all $\sigma_\varepsilon$ values are below
1.07 mm/day due to the overall high consistency among ET products, with correlations between $0.39 < \rho_{t,X} < 0.99$. However,
products with highest $\rho_{t,X}$ necessarily do not have the lowest $\sigma_\varepsilon$. Hence, the discrepancy between products varies but does not
dominate differences in the sensitivity among products. The highest $\sigma_\varepsilon$ is found at station FR-Bil for GLDAS-2, when
comparing GLDAS-2 with GLEAM and ICOS. The lowest $\rho_{t,X}$ of 0.33 is found at station DE-Ruw for ICOS as the results of
the ETC among GLDAS, MODIS, and ICOS. Despite the high ECC values at DE-Rus (see Fig. S8) and hence, potential
product dependencies, ETC results at this station are inconspicuous with comparable errors and correlations. Overall, ERA5-
land, SEVIRI, and GLEAM perform better at all stations with either lowest errors or highest correlations within their ETC
triplets. In summary, compared to ERA5-land and ICOS, the remote sensing products (SEVIRI, MODIS, GLEAM) show
similar uncertainties as ERA5-land, but at most stations ERA5-land outperforms GLEAM and MODIS (see Fig. 6, upper row).
Further, compared to GLDAS-2 and ICOS, the remote sensing products in most cases outperform GLDAS-2 and ICOS,
showing the lowest uncertainties, i.e. lower errors and higher correlations (see Fig. 6, lower row). During all analyses, ICOS
shows generally the highest uncertainties. Potential explanation is the discrepancy in spatial resolutions (see Tab. 1) as will be
discussed in more detail in sec. 4.

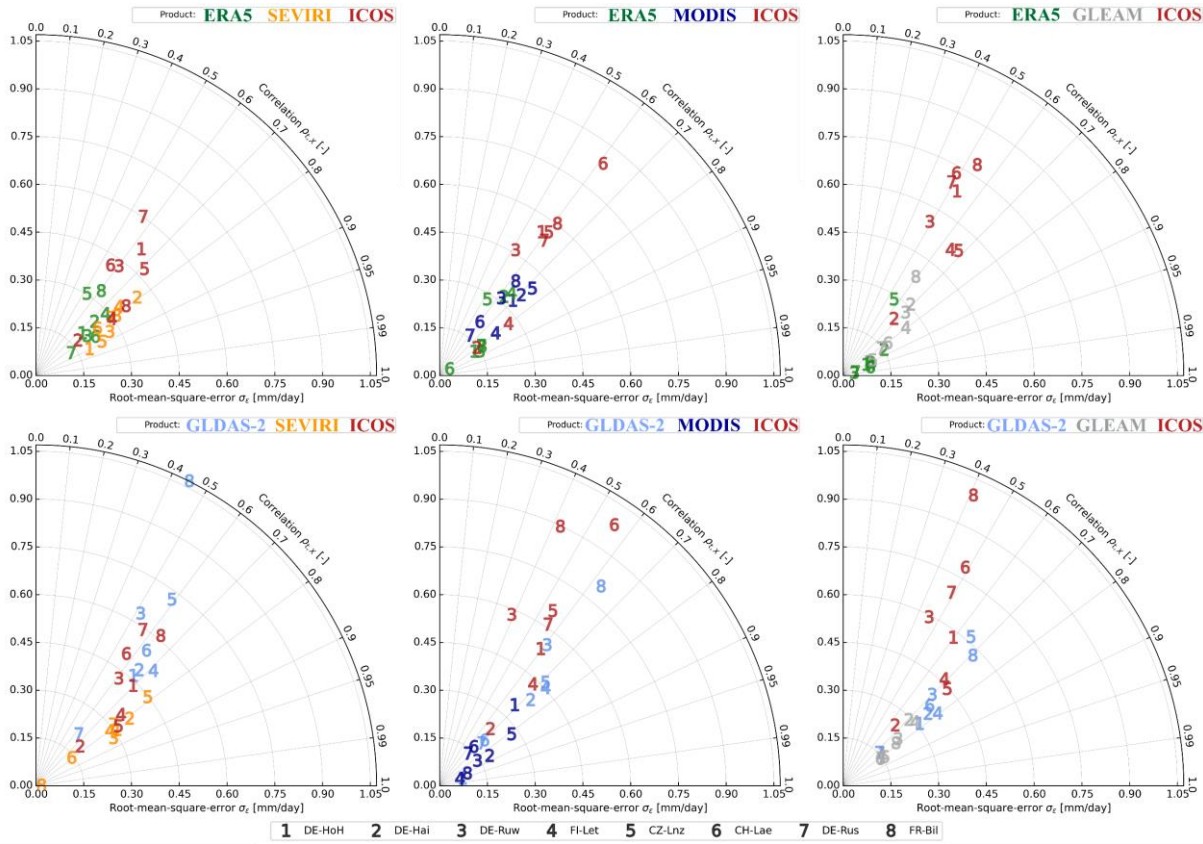


Figure 6: Estimated root-mean-square-error ($\sigma_\varepsilon$) [mm/day] (on the x- and y- axes) and correlation ($\rho_{t,X}$) [-] (on the arcs) among ET products
at all stations based on the extended triple collocation (ETC) method from McColl et al., (2014). Numbers represent the eight stations and
colours the different ET products. 1st row: ETC between SEVIRI, MODIS, and GLEAM datasets respectively with ERA5-land and ICOS.
2nd row: ETC between SEVIRI, MODIS, and GLEAM datasets respectively with GLDAS-2 and ICOS.

## 3.2 Drought impacts on ET products

As shown in Figures 3 and S1, 2018 was an exceptional dry year across central Europe. In this section, the impact of the
drought in 2018 on ET is investigated by comparing it to SM and VPD, the two main parameters that are used for monitoring
drought-related terrestrial ecosystem productivity (see Sec. 1). For that, we will compare 2018 always to the rather wet year
2017 to identify significant changes.
In Figure 7, the time series of ICOS ET, SMAP SM, and in-situ measured VPD for 2017 and 2018 are compared to their
respective calculated anomalies (see Sec. 2.3.2) for DBF (DE-HoH, DE-Hai) and ENF (DE-Ruw, FI-Let) stations. While ET
and VPD show a distinct seasonal pattern at all stations with highest values during summer months, SM shows a less clear
seasonal pattern with more inter- and intra-annual variations. At both DBF stations and the ENF station DE-Ruw, the highest
SM values are generally found during the winter months. In contrast, at ENF station FI-Let, an almost constantly increasing
SM in 2017 can be observed with a distinct drop from in January 2018 and subsequent distinct increase in April 2018. The SM
also stays at high values throughout the entire summer until mid of October in 2018, besides a smaller decrease from end of
May until August. However, these SM values may be an artefact of snow cover or frozen ground at the northernmost station
and should be treated carefully, although the MT-DCA is quality controlled and filtered for that (see Sec. 2.2).

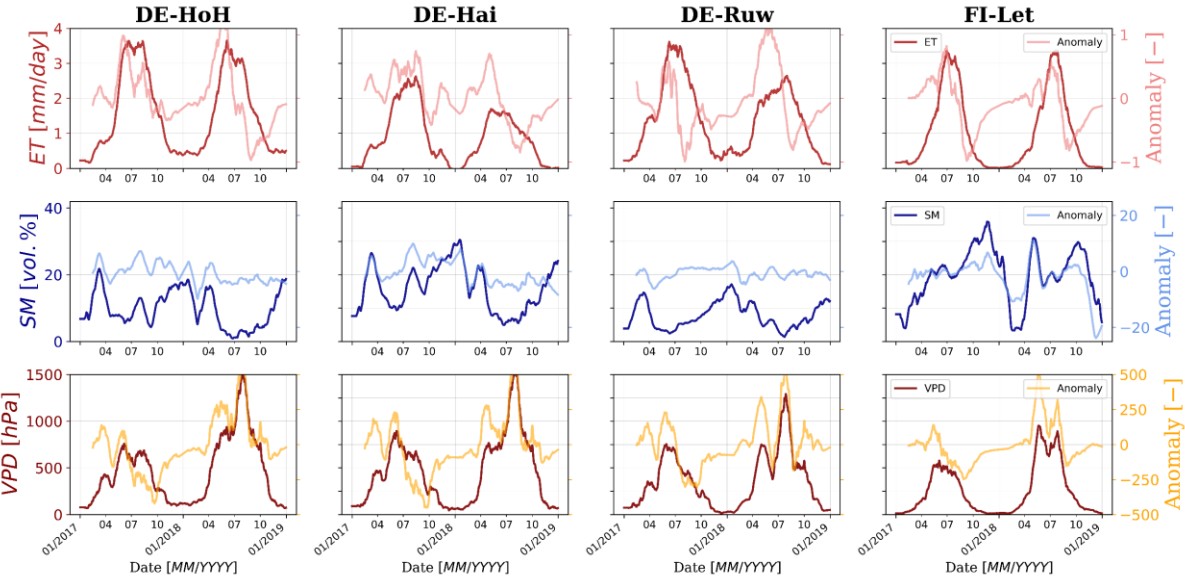

Figure 7: Time series of daily ICOS ET [mm/day], SMAP SM [vol.%], and in-situ VPD [hPa] for 2017 and 2018 at DBF (DE-HoH, DE-
Hai), and ENF (DE-Ruw, FI-Let) stations compared to their respective anomalies (see Sec. 2.3.2). All time series were cleaned for daily and
weekly dynamics using a Savitzky-Golay (Savitzky and Golay, 1964) filter with a window size of 31 days.
From these time series, in general lower ET and higher VPD values can be found in 2018 compared to 2017, reflecting the
drought conditions with higher atmospheric aridity and decreased water supply for plant transpiration and soil evaporation in
2018. At the MF (CZ-Lnz, CH-Lae) and agriculture (DE-Rus, FR-Bil) stations, the same trends can be observed but with minor
differences in VPD maxima between 2017 and 2018, and sometimes higher ET peaks in 2018 at stations CZ-Lnz and FR-Bil
(see Fig. S9). The overall lowest SM values can also be found in 2018, except at station FI-Let. At the DBF stations and station
DE-Ruw, constantly low SM values over several months from mid of April to mid of October are shown without any significant
increase during this time in 2018 (see Fig. 7). The same is true at MF station CH-Lae and the agricultural stations. At station
CZ-Lnz, SM is varying monthly at low values between ~5 vol.% and 18.6 vol.% (see Fig. S9). When analysing the anomaly
time series (seasonal detrending; see Sec. 2.3.2) of each parameter and station, in general higher ET and VPD anomalies and
lower SM anomalies are found in 2018 compared to 2017, except at station FI-Let with higher SM anomalies in 2018 compared
to 2017 (see Figs. 7 & S9).
These anomalies are subsequently used in Figure 8 to visualize the kernel densities of SM, VPD, and ET anomalies of all
stations for 2017 and 2018. In Figure 8, only the vegetation periods from April to October within each year are analysed. It
can be seen that in 2018 (drought year), the SM and ET anomalies peak at lower, negative values compared to 2017, where
they peak around zero, while the VPD anomalies peak at higher, positive values compared to 2017. Also, the respective
anomaly medians are lower for SM and ET, and higher for VPD in 2018. The calculated $p$-values of always $\leq 0.045$ prove the
shift in yearly median values at the 5 % significance level.

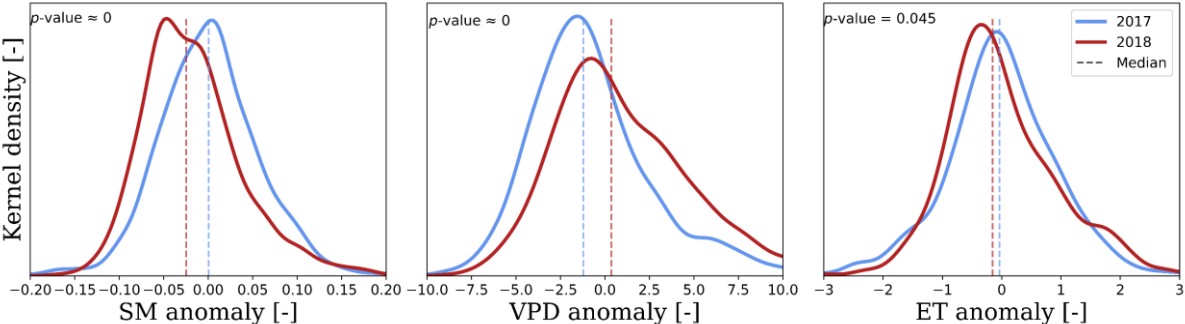

Figure 8: Kernel density estimates of daily SMAP SM, in-situ VPD, and ICOS ET anomalies (see Sec. 2.3.2) during April to October of
2017 and 2018 across all investigated stations. The dashed lines represent the seasonal median of respective parameters and years. The **$p$-**
values of a two-sided Wilcoxon rank-sum test indicate the acceptance ($> 0.05$) or rejection ($< 0.05$) of the null hypothesis regarding
continuous distributions with equal medians at the 5 % significance level.
When comparing the anomalies for different ET products (see Fig. 9), a similar shift towards lower values for 2018 compared
to 2017 can be found for MODIS and ERA5-land products. For SEVIRI, GLDAS-2, and GLEAM a shift towards higher
anomalies in 2018 is found with medians at slightly higher values compared to 2017. However, while the ICOS $p$-value of
0.045 being close to the 5 % significance level of equal medians, the ones of SEVIRI, GLDAS-2 and GLEAM are more
significant around zero. GLEAM anomalies peak at the same value for both years but with higher positive anomalies for 2018
at values greater than 0.6. In general, Gaussian distributions around zero are evident for both years at all anomalies of ET
products. Only at MODIS, a clear bimodal distribution in ET anomalies of 2018 with a first peak around -0.4 and a subsequent
second smaller peak at 0.55 can be found. This is also the ET product with the smallest anomaly range from -1.5 to 2.5. All
other ET products vary at least between -3 and 3. For the ET products ERA5-land, GLDAS-2, and GLEAM, a non-linear
decrease in 2018 can be found with almost stagnating anomalies around one. For the ICOS and SEVIRI data, this trend is first
visible at values greater than one. In contrast, the density curves of ET anomalies for 2017 are smoother for all products,
showing a clear Gaussian distribution. Again, the calculated $p$-values of $\leq 0.02$ prove the shift in yearly median values at the
5 % significance level, except for the MODIS product ($p$-value $< 0.1$). The MODIS product is also the ET product with the
lowest temporal resolution of eight days (see Tab. 1). When analysing all other ET products at the same 8-daily resolution (see
Fig. S10) similar bimodal distributions in 2018 can be found for ERA5-land, SEVIRI, and GLEAM. GLDAS-2 shows even a
trimodal distribution with the highest density of ET anomalies around -4.5, a second peak around 1.4, and a third peak around
6.3. Although no clear bimodal distribution can be seen for ICOS even at 8-daily resolution, the distribution smoothly increases
from -15 to -4 and then non-linearly decreases with at least three smaller plateaus (see Fig. S10). And even for 2017, the
Gaussian distributions are not that smooth as for the daily analyses. More detailed analyses revealed that there is a distinct
drop in 8-daily anomaly time series, leading to this bimodal distribution. Between April and August almost only positive ET
anomalies are found, while during September and October almost only negative anomalies are found. The same trend is, of
course, also visible for the daily time series but due to the preserved daily and intra-weekly dynamics, the difference between
positive and negative anomalies during both periods (April-August, September-October) is not that distinct. These small-scale
dynamics are excluded in the 8-daily analyses. However, the differences in ET anomalies between 2017 and 2018 are greater
for the 8-daily anomaly analyses (see Fig. S10) compared to the daily anomaly analyses (see Fig. 9), indicating that drought
impacts on ET are more pronounced at larger time scales (more than a week, monthly) than on smaller time scales (less than
a week, daily). In summary, the reason for the bimodal distribution in ET anomalies within the MODIS products originates
from the lower temporal resolution.

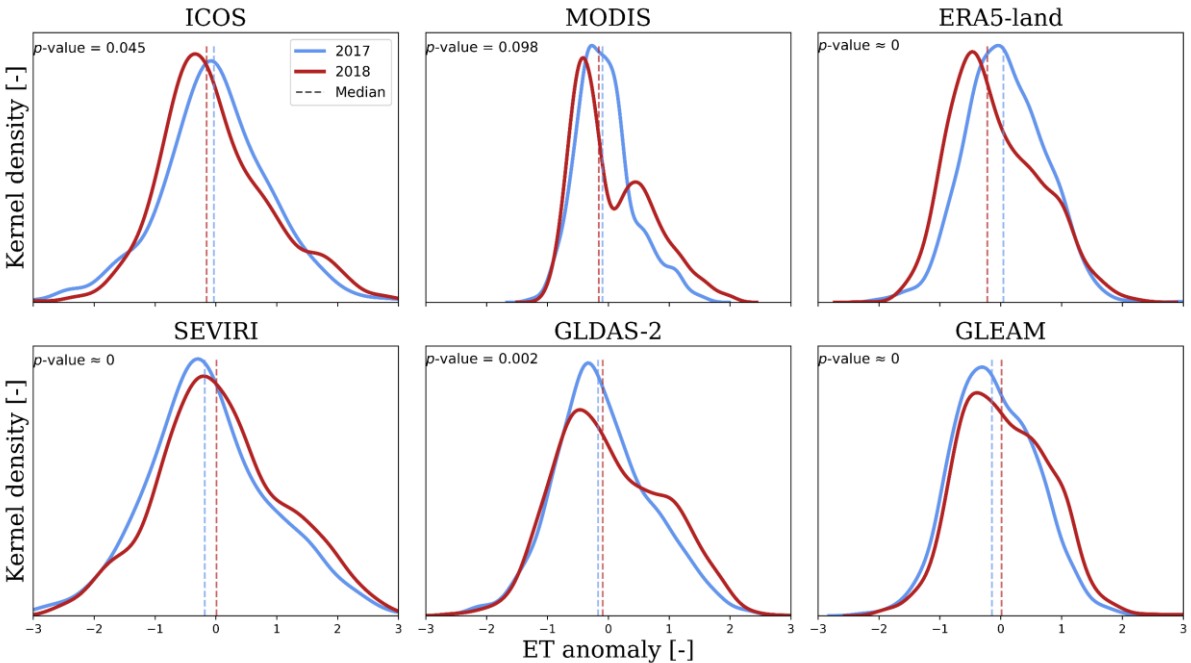

Figure 9: Kernel density estimates of daily ET anomalies (see Sec. 2.3.2) for all investigated ET products during April to October of 2017
and 2018 across all investigated stations. The dashed lines represent the seasonal median of respective parameters and years. The $\boldsymbol{p}$-values
of a two-sided Wilcoxon rank-sum test indicate the acceptance ($> 0.05$) or rejection ($< 0.05$) of the null hypothesis regarding continuous
distributions with equal medians at the 5 % significance level.
For analysing the dependencies between ET, SM and VPD, respective ET products in SMAP SM and in-situ measured VPD
bins (see Sec. 2.3.3) are visualized for the wet year 2017 (see Fig. 10) and the dry year 2018 (see Fig. 11) across all stations.
ET for all stations and both years are similarly distributed across the SM and VPD phase space.
For the rather wet year 2017, a general decreasing trend in ET values along increasing VPD and increasing SM can be found
for all ET products except SEVIRI. Here, a decreasing trend along increasing VPD but decreasing SM is visible as indicated
by the arrow within the inset plot (see Fig. 10). Overall, ET varies more with VPD than SM. Only ET from ICOS and to some
extend ERA5-land and GLEAM have highest values at intermediate VPD and SM, and lower ET at low SM. Especially ET
products SEVIRI and GLDAS-2 do not display any reductions at low SM.

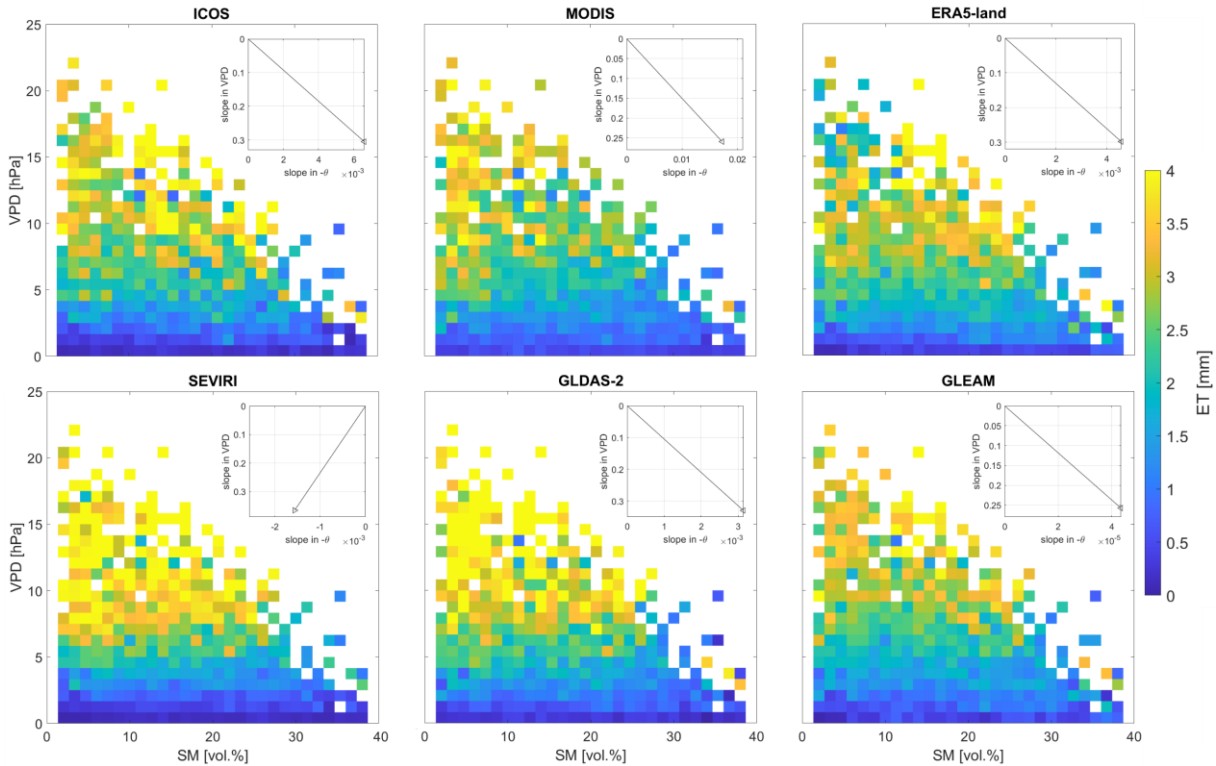


Figure 10: ET [mm] relative to SMAP SM [vol.%] and in-situ VPD [hPa] for all investigated ET products and averaged over all investigated
ICOS stations in 2017. The inset plots provide the corresponding median slope in SM and VPD changes.
For the dry year 2018, only MODIS and GLDAS-2 still show a decreasing trend along increasing VPD for increasing SM. All
other products indicate decreasing ET for increasing VPD and decreasing SM (see. Fig. 11). At SEVIRI, the slope in SM
direction is twice as low in 2018 compared to 2017 but almost the same for VPD, meaning greater decrease in ET along SM
during the dry year. A similar trend is observable at MODIS with half of the slope along SM in 2018 compared to 2017,
meaning half as strong increase in ET values with SM during the drought affected year 2018. Lastly, at GLDAS-2, the slope
along SM bins is increased by a factor of almost seven in addition to a reduced slope in VPD of ~0.1 hPa in 2018, meaning
stronger increase in ET values at increasing SM at simultaneously decreasing VPD during the drought year. Further, ET values
are in general lower in 2018 compared to 2017, but in 2018, bins at higher VPD values with low ET can be found across the
entire SM range (see Fig. 11).
In summary, for both years, ET is generally higher at high VPD, i.e., higher atmospheric water demand, and much lower below
a VPD of 5 hPa. In Figures 10 and 11, we do not really see very clear reductions of ET with decreasing SM. Hence, ET varies
more with VPD than SM. The influence of SM on ET is only noticeable when comparing the wet (2017) and dry (2018) years
with each other, as the change along SM ($\frac{\Delta ET}{\Delta SM}$) is significantly higher during the drought affected year.

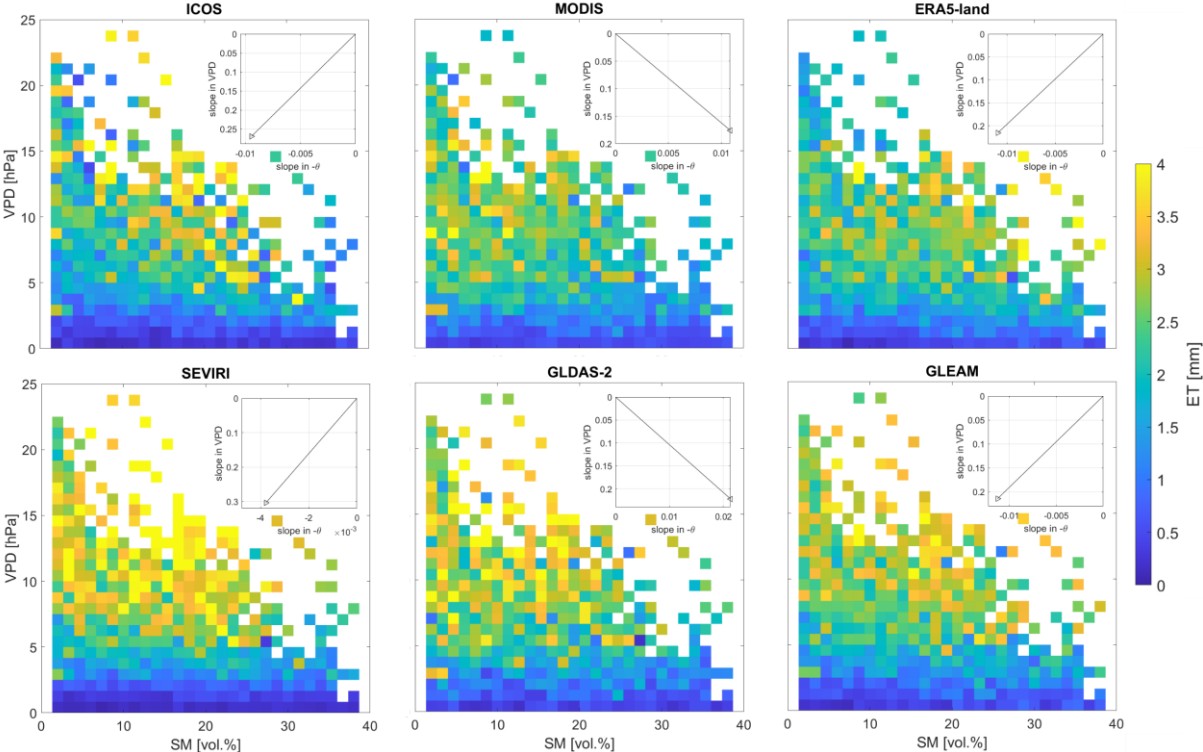


Figure 11: ET [mm] relative to SMAP SM [vol.%] and in-situ VPD [hPa] for all investigated ET products and averaged over all investigated
ICOS stations in 2018. The inset plots provide the corresponding median slope in SM and VPD changes.
For more detailed analyses regarding the drought effect on ET products, we calculated the coefficient of variation (CV) [%]
for 2017 and 2018 (see Fig. 12) for each ET bin relative to the SM and VPD ranges (see Sec. 2.3.3.). CV is defined as ratio
between the standard deviation and the mean and provides the relative dispersion or amount of uncertainty of data. As can be
seen, the differences in CV between 2017 and 2018 are highest for low SM. Here, the variability in ET values during the
drought year of 2018 reach higher VPD values compared to 2017. Furthermore, overall higher CV are estimated for low VPD
across the entire SM range in 2018 compared to 2017. In contrast, 2017 shows slightly higher CV values for intermediate
values (SM between 10 and 30 vol.% and VPD between 4 and 8 hPa). When comparing the different ET products among each
other, the CV show overall similar patterns. However, for both years the CV median for ICOS (49.23 %, 48.43 %) are always
the highest compared to all other products, indicating a greater dispersion of data points within the ET time series. Lowest
median CV are found within ERA5-land with 33.28 % in 2017 and 36.48 % in 2018 (see Fig. 12), indicating overall lowest
amount of uncertainty in ET data.

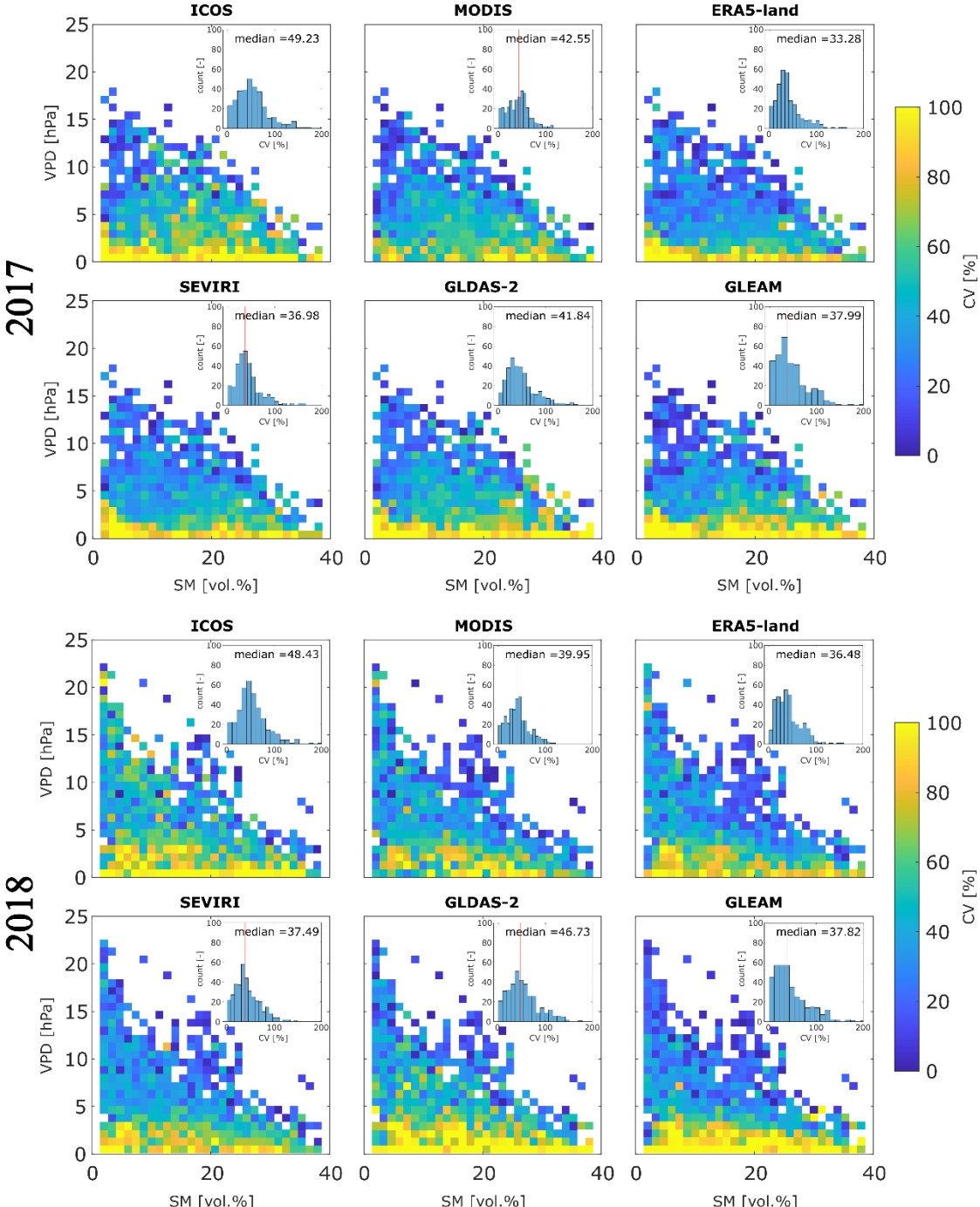

Figure 12. Coefficient of variation (CV) [%] of ET values relative to SMAP SM [vol.%] and in-situ VPD [hPa] for all investigated ET products and averaged over all investigated ICOS stations for 2017 (upper two rows) and 2018 (lower two rows). Inlet plots give the histogram of displayed CV values with the red line indicating the location of the given median for each product.


## 4 Discussion

### 4.1 Differences in examined ET products

When evaluating the performance of all ET products from remote sensing, reanalysis, modelling and ground-based eddy covariance measurements, analyses of their time series revealed that the ICOS ET almost always show a time lag of about few weeks during spring ET rise compared to all other products (see Fig. 4). This could be explained by the discrepancy in spatial resolutions, with the ICOS product providing local point-scale measurements compared to the other larger-scale remote sensing and modelling ET products. This spatial mismatch alters the vegetation impact within the ET signal. Another reason is the dependency of models on indicators for phenological changes in vegetation. For example, many models use the leaf-area index (LAI) to track phenology dynamics, which influence ET simulations (Adeluyi et al., 2021). Further, the overall lowest ET values were found for all products at the agricultural station DE-Rus, while highest values were found at the southernmost station FR-Bil, where the highest average precipitation was recorded between 2017 to 2020 (see Fig. 3). Reason for that are for one, reduced transpiration of agricultural sites throughout the year compared to forested sites, and second, the humid Atlantic climate at the southernmost station at lowest altitude (see Tab. S1). Further, the mostly non-irrigated arable land at station DE-Rus (see Fig. 2) shows relatively low vegetation cover (LAI < 2; normalized difference vegetation index (NDVI) around 0.5 during summer months (not shown)) compared to forested sites (LAI > 5, NDVI around 0.8 during summer months (not shown)), which can lead to an underestimation of ET when using models that rely on vegetation indices (i.e., NDVI, LAI). Combined with the seasonal vegetation dynamics of this station and the lack of irrigation, this explains the lower ET values compared to forested areas with more consistent canopy cover. The 8-day analyses showed that MODIS gives higher values compared to all other ET products at two stations, while ICOS is higher than all other ET products at one station. Further, the highest density of values was found between 0 to 10 mm/8-days due to the seasonal imprint with reduced ET across Europe during months with reduced solar radiation and vegetation cover (November-March). Only at the two coniferous forest stations (DE-Ruw, FR-Bil), the highest density of values is between 0 to 20 mm/8-days with lower ET values only during winter months (December-February). However, this does not apply to the third coniferous station FI-Let, which is the northernmost station with less dense forests and more snow fall between November and March, which influences the estimation of ET. Hence, the lack of leave-off conditions and the reduced amount of days with snow cover influences the amount of ET. Conducted statistics confirmed the noticeable differences among ET products and ICOS stations, which indicated an overall lower agreement among products at the rather mixed agricultural station (DE-Rus) and better consistency at ENF stations (DE-Ruw, FI-Let). Hence, products differ most at stations with complex land cover conditions, where varying crops and growing seasons (changing phenology) make the estimation of ET more difficult, while evergreen needle-leaved stations with less changes throughout the year and between years are easier to define (temporal homogeneity).

For more detailed product performance analyses, the extended triple collocation (ETC) method (McColl et al., 2014) and SM-VPD binning revealed highest uncertainties (see Fig. 6) and coefficient of variation (see Fig. 12) for the ICOS product, and lowest uncertainties for SEVIRI and GLEAM as well as ERA5-land. The highest ETC error was estimated for GLDAS-2,

when analysing with GLEAM and ICOS, while the lowest sensitivity (correlation) was found for ICOS, when analysing with GLDAS-2 and MODIS (see Sec. 4.1). Hence, the remote sensing products (SEVIRI, GLEAM) and the reanalysis product (ERA5-land) differed most from the in-situ field-scale (ICOS) and modelling (GLDAS-2) products. One reason for the mismatch between the ICOS product and SEVIRI, GLEAM and ERA5-land is surely the spatial mismatch between the point-scale ground-based EC tower measurements and the remote sensing (3 km) or reanalysis (9 km) products. However, in order to capture vegetation stress, ecosystem health, and fine-scale variability in ET globally, adequate spatial (and temporal) resolutions are necessary. Here, detailed research regarding downscaling techniques, as reviewed in, e.g. (Mahour et al., 2017; Peng et al., 2017), that combine medium-scale ET data with fine-scale auxiliaries in order to improve the spatial resolution, are needed regarding its uncertainties and impact on product comparisons. Further, ET measurements based on the eddy covariance method tend to underestimate sensible heat (H) and latent heat (LE) fluxes (Petropoulos et al., 2015), are often temporally too short and spatially too sparse to sample drought conditions correctly (Zhao et al., 2022), and suffer from challenges to close the energy balance (Yu et al., 2023). Several studies (Twine et al., 2000; Petropoulos et al., 2015; Barrios et al., 2024) reported an error range of EC measurements of ~10-30 % due to, e.g., a 'systematic closure problem in the surface energy budget' (Twine et al., 2000). In order to identify potential product dependencies, which may impact the ETC results, the estimated error cross-correlations (ECC) were calculated, with high ECC between GLDAS-2 and GLEAM (at DE-HoH), between ERA5-land and GLEAM (at CZ-Lnz), and all products and GLEAM (at DE-Rus). These need to be accounted for when analysing the differences among ET products. Although in this study, we have analysed different land cover classes within a 3 km footprint around every ICOS station at daily resolution to account for the different resolutions, the SEVIRI product provides ET data every 30 minutes at moderate spatial resolution (3 km), and showed to capture ET dynamics on small as well larger temporal scales comparable or even better than other examined products, as also reported by previous studies, e.g., (Hu et al., 2015; Petropoulos et al., 2015; De Santis et al., 2022). None of the other examined products can provide similar spatio-temporal coverage, due to either lower temporal resolution (MODIS) or coarser spatial resolution (ERA5-land, GLDAS-2, GLEAM). Only the ICOS data provide similar temporal resolution to SEVIRI but at point-scale, which disqualifies it for global analyses. Although there exist other ET products from remote sensing and modelling, e.g., (Jiménez et al., 2011; Mueller et al., 2013; Fisher et al., 2020; De Santis et al., 2022; Yu et al., 2023), the examined ET products in this study are appropriate when addressing global analyses since other products have either a more coarse spatial or temporal resolution (Yu et al., 2023), are limited to clear sky conditions (De Santis et al., 2022), which prohibits continuous time series of ET measurements, or are higher order derivates from either field measured or merged remote sensing based products (Jung et al., 2019; Chen et al., 2021). We also analysed data from the ECOsystem Spaceborne Thermal Radiometer Experiment on Space Station (ECOSTRESS) launched by NASA in June 2018 (Fisher et al., 2020) at the beginning of our analyses. However, we found several problems with this product and worse performance compared to other ET products, meaning a clear overestimation using the ECO3ETPTJPL product, as reported also by previous studies, e.g., (Liu et al., 2021; De Santis et al., 2022; Wu et al., 2022). In our research with ECOSTRESS, data was unavailable at CZ-Lnz and FI-Let. Another ECOSTRESS ET product, the ECO3ETALEX (based on the DisALEXI model), has shown better performance, but it is more suited for agricultural

applications, and it is limited to the United States (Cawse-Nicholson and Anderson, 2021). ECOSTRESS level 3 ET data come
at the advantage of a high spatial resolution (70 m), but its temporal resolution is irregular due to the ISS orbit and the
dependency on the product type and study region limited our preliminary analyses. For these reasons, we decided not to include
it in our research.

## 548 4.2 Impact of droughts on ET products

Since remote sensing-based ET products are not purely observational, the performance of an ET product is highly dependent
on the employed retrieval model for ET estimation. This is in turn dependent on how the model deals with limitations in SM
or VPD and responses under drought conditions. Every retrieval method has its own strengths and weaknesses, but especially
under drought conditions, the ability of the employed algorithm to deal with water shortage and vegetation stress is essential
for valid ET estimation. Varying types of vegetation have different strategies how to deal with water stress, e.g., by closing
stomata to prevent water loss through leaves and increasing the water uptake from the soil or deeper soil depths by increasing
the water resistance (He et al., 2022). Many studies reported decreasing ET during droughts due to reduced SM supply and
hence, decreasing evaporation, but also decreasing transpiration since plants close their stomata to prevent water loss (Novick
et al., 2016; Zhao et al., 2022). However, during drought conditions with increasing air temperatures, ET can also increase due
to the higher atmospheric moisture demand (increasing VPD). Further, the generic statement that ET decreases due to
decreasing SM often ignores the fact that plants have access to SM from greater soil depths, which are not immediately affected
by meteorological droughts, or have different strategies for drought resistance (Gupta et al., 2020; He et al., 2022; Feldman et
al., 2024). Hence, the dynamics of ET to drought conditions remain highly variable (Zhao et al., 2022). Novick et al., (2016)
pointed out that SM and VPD may become more decoupled in the future and models need to resolve limitations in SM and
VPD independently from each other in order to capture the response of ecosystems to water stress correctly (Novick et al.,
2016; Zhao et al., 2022). How models react to limitations in SM and VPD varies significantly which impacts resulting ET.
Analyses performed in this study revealed that during the rather wet year 2017, ET varied more with VPD than with SM, with
almost no dependency of ET on SM in SEVIRI and GLDAS-2 products. Here, our results indicate that ET is more controlled
by atmospheric demand rather than water supply from atmosphere (precipitation) and soil (soil moisture) as reported also by
Zhou et al., (2019). However, it is suggested by previous work and the Budyko framework (Budyko and Miller, 1974) that ET
should exhibit some level of dependence on SM (Porporato et al., 2002; Zhang et al., 2021). One reason could be that forests
at selected ICOS stations might have substantial access to deeper SM (root zone) that exceeds the measurement depths of the
SMAP satellite (first 25 cm) (Feldman et al., 2022). When analysing the controls of SM and VPD on ET during the dry year
2018 however, all ET products, except MODIS and GLDAS-2, showed that ET decreases with increasing VPD and decreasing
SM. For SEVIRI, even a twice as large decrease in ET along SM during the drought year could be observed compared to the
rather wet year. This declining trend of ET during dry years when ET is limited by moisture and VPD is increasing due to
increasing air temperatures is in line with previous studies (Jung et al., 2010; Seneviratne et al., 2010; Zhou et al., 2019).
Further, results show that VPD and SM are negatively coupled during extreme events as reported also by (Zhou et al., 2019;
De Santis et al., 2022). However, MODIS and GLDAS-2 products showed an increase of ET with increasing SM and with
decreasing VPD during 2018 (see Fig. 11). These are the two products that are based on the Penman-Monteith equation (see
Tab. 1), and that were outperformed by SEVIRI, ERA5-land and GLEAM in the ETC analyses (see Fig. 6). For MODIS, one
reason for the worse performance was found to be the coarse temporal resolution of 8-days, since at this time scale the temporal
variability of ET is significantly different lacking all diurnal and day-to-day ET dynamics. The underperformance of MODIS
compared to in-situ EC measurements was also reported by (De Santis et al., 2022), who found that MODIS overestimated in-
situ ET measurements at stations in Italy, as well as (Yu et al., 2023), who investigated several stations with different land
covers and varying climatic zones across the U.S. They concluded that daily or monthly ET products performed best compared
to EC tower measurements (Yu et al., 2023). Due to the temporal resolution, MODIS is the only product showing a bimodal
distribution of ET anomalies with a $p$-value above the 5 % significance level (see Fig. 9). In this study, we could show that
differences in ET anomalies between 2017 and 2018 are greater for the 8-daily anomaly analyses (see Fig. S10) compared to
the daily anomaly analyses (see Fig. 9), indicating that drought impacts on ET are more pronounced at larger time scales (more
than a week, monthly) than on smaller time scales (daily, less than a week). Hence, the temporal scale for ET analyses is
crucial in order to select which temporal component of the ET dynamics should be considered for a respective application.
Further, although GLEAM is built on the less parameterized Priestly-Taylor equation compared to the Penman-Monteith
equation since it does not consider VPD or canopy conductance on soil water stress, the GLEAM ET product showed to deliver
better ETC results and statistics in this study. A comparable or even better performance of the Priestley-Taylor equation
compared to the Penman-Monteith was also reported in previous studies, e.g., (Akumaga and Alderman, 2019; Bottazzi et al.,
2021). Reasons could be the uncertainties of input variables within the Penman-Monteith equation, e.g., for stomatal, canopy,
or aerodynamic resistances, which are often unknown, approximated (Widmoser, 2009), or parameterized based on the wrong
variable (Hu et al., 2015), or due to the overestimation of specific parameters, such as the net radiation, or other aerodynamic
factors as reported by (Hao et al., 2018). Similar, Hu et al., (2015) stated that MODIS tends to overestimate water stress during
thawing of frozen soil in Spring or over irrigated land, which leads to an underestimation of soil evaporation. Moreover, several
studies pointed out that the Penman-Monteith equation needs to be adapted for climate/weather extremes and vegetation limited
cases, e.g., (Widmoser, 2009; Hao et al., 2018; McColl, 2020).
The estimated coefficient of variation (CV) showed that during the drought year 2018 ET values display highest uncertainty
for low SM or low VPD, while during the rather wet year 2017, the highest variability was found for intermediate SM and
VPD values. Hence, our results show that during drought conditions the estimation of ET leads to highest uncertainties and is
most difficult for low SM and low VPD depending on the assumptions for controlling factors on ET. For example, within the
Penman-Monteith equation, aerodynamic as well as stomatal resistances are considered but since they can vary significantly
for drought and non-drought conditions erroneous assumptions for them can lead to significant errors. During normal or wet
conditions, as shown for 2017, CV results do not vary that much between investigated ET products but indicate for all highest
variability for intermediate values of SM and VPD, which originate most probably from different vegetation and ecosystem

types. Here, more research for individual ecosystem types is required to further address ET controls, as vegetation is the controlling factor on ET estimates when SM and VPD are not limited (Brown et al., 2010; Jin et al., 2017).

In summary, it is important for ET retrieval algorithms to account for water droughts and vegetation stress as done with adaptable stomatal closure and canopy resistance within the Penman-Monteith equation. However, analyses showed, that false assumptions on these physiological stress indicators can decrease the performance and be exceeded by less parameterized and simpler retrieval algorithms like the Priestley-Taylor equation.

**5 Conclusion and Outlook**

In this study, eight different ET products with varying temporal and spatial resolutions as well as varying ET retrieval methods (see Tab. 1) are analysed across central Europe for the period of 2017 to 2020. Despite the spatial mismatch (in-situ vs. remote sensing) and the spatial heterogeneity of the analysed landscapes (see Fig. 2), all products showed a concurrent seasonal pattern and overall low uncertainties during ETC analyses. It was shown that ET varied from year to year for different forest and agricultural stations due to changing seasonal weather and vegetation conditions over the years. Analyses revealed that temporal and spatial homogeneity helps with the consistency and interpretability of the ET estimates. This is, since products were most consistent with each other at stations with less complex land cover conditions and changes throughout the seasons (the evergreen needle-leaved stations DE-Ruw and FI-Let). Despite the good match in seasonal patterns, differences in ET products were noticeable. The remote sensing products, SEVIRI, MODIS, and GLEAM, performed equivalently well or even better than the in-situ measured (ICOS), modelled (GLDAS-2) or reanalysis (ERA5-land) products for this specific study concept (3 km footprint, daily analyses). Extended triple collocation (ETC) and SM-VPD binned ET analyses revealed that SEVIRI and ERA5-land (the two products based on the (H-) Tessel land surface scheme) perform best. They provide low uncertainties when compared with other products and reasonable SM and VPD controls on absolute ET. GLEAM also shows a good performance, although this result should be taken with caution since potential product dependencies with ERA5-land and GLDAS-2 may have affected the ETC results. When analysing the behaviour of ET in context of SM and VPD during the rather wet year 2017 and dry year 2018, it was found that in 2017, ET is highly dependent on VPD and less on SM. Hence, with sufficient moisture supply, ET is mainly controlled by atmospheric demand and the vegetation transpiration. In contrast, in 2018, limited moisture supply because of decreasing SM and increasing VPD, which were in turn due to increasing air temperatures, led to a decline in ET, in line with previous studies. Further, during the dry year 2018, SM and VPD were more negatively coupled which could also had an impact on the ET decline. These behaviours were consistently found in all ET products, except for GLDAS-2 and MODIS, the two products whose retrieval methods are based on the Penman-Monteith equation. Hence, although GLEAM is based on the less parameterized Priestley-Taylor equation compared to the Penman-Monteith equation, it is outperforming GLDAS-2 and MODIS within this study set-up, which supports the idea to adapt the Penman-Monteith equation as reported by previous studies, e.g., (Widmoser, 2009; Hao et al., 2018; Akumaga and Alderman, 2019; McColl, 2020; Bottazzi et al., 2021).

Further, the comparison between estimated coefficient of variation (CV) for 2017 and 2018 showed that the dispersion of data is higher during the extreme drought year 2018 for extreme conditions, such as low SM or low VPD across all SM values. In contrast, 2017 showed higher CV for intermediate conditions. However, the difference between investigated products is rather minor, with median CV between 33.28 % (ERA-land) and 49.23 % (ICOS), and should be analysed in future studies for individual stations and ecosystem types (requiring longer time series and more stations to have enough data points for binning) for determining the impact of varying vegetation types on ET controls. In summary, when considering all conducted analyses together (spatial and temporal resolutions, product dependencies, ETC results, SM and VPD controls on ET), the remote sensing products SEVIRI and GLEAM as well as reanalysis product ERA5-land seems to provide most reasonable results compared all other ET products, with SEVIRI providing a higher temporal and spatial resolution compared to GLEAM and ERA5-land. Hence, despite their coarse spatial resolution, GLEAM and ERA5-land are able to capture ET dynamics sufficiently even under drought conditions. Future research regarding data fusion techniques and downscaling approaches that combine coarse- or medium-scale ET data with fine-scale auxiliaries in order to improve the spatial resolution of certain ET products may help to decrease the spatial mismatch and optimize the comparison between point-scale field measurements and satellite remote sensing or modelling data.

This study served as a pathfinder to compare freely available and commonly employed ET products at highly monitored EC towers across central Europe. Whether these reported findings hold true across space and for other drought events has to be analysed further with focus on spatially larger regions and longer time series. Additionally, potential add-on studies could include the examination and comparison of ET dynamics from optical/thermal remote sensing observations with microwave remote sensing data, e.g, the Sentinel-1 backscatter, in order to evaluate the potential of active microwave remote sensing for drought monitoring, e.g., (Mueller et al., 2022; Jagdhuber et al., 2023). In order to identify relevant conditions and causal strengths with lagged and contemporaneous causal dependencies between different variables, like ET, the Sentinel-1 backscatter and other important SPAS parameters, like air temperature, relative humidity, and water potentials, the use of emerging powerful tools for causal discovery could prove useful (Runge et al., 2019; Díaz et al., 2022). Previous studies already outlined the potential of identifying causal relations between Earth system parameters (i.e., precipitation, ET, SM, air temperature) by using the wavelet coherency analysis (WCA) (Graf et al., 2014; Rahmati et al., 2020), or the PC algorithm Momentary Conditional Independence (PCMCI) method (Runge et al., 2019, 2023).

*Data availability.*

The SMAP MT-DCA V5 soil moisture dataset is available at https://zenodo.org/records/5619583, last access: 11 May 2022. The SPEI dataset is available at https://spei.csic.es/database.html, last access: 18 November 2023. The evapotranspiration products are available as follows: ICOS data are available at https://www.icos-cp.eu/, last access: 20 November 2023. SEVIRI data are available at https://datalsasaf.lsasvcs.ipma.pt/PRODUCTS/MSG/MDMETv3/, last access: 21 November 2023. MODIS data are available at https://lpdaac.usgs.gov/products/mod16a2v061/, last access: 20 November 2023. ERA5-land data

are available at https://cds.climate.copernicus.eu/datasets/reanalysis-era5-land?tab=overview, last access: 20 November 2023.

The GLDAS-2 data are at https://ldas.gsfc.nasa.gov/gldas/model-output, last access: 22 November 2023. The GLEAM data are available at https://www.gleam.eu/, last access: 23 August 2024. The Corine land cover classes are available at https://land.copernicus.eu/en/products/corine-land-cover/clc2018?hash=4ecde146e6ca8dd7a42f68a9f5370153d9731a95, last access: 14 March 2024.

*Author contributions.*

TJ designed the study concept and assembled the research team. AF, MB, MP, BB, MR, CM, and TJ were involved in the data acquisition and in developing the methodology. AF led the data curation and visualization of results. The original draft was written and prepared by AF and TJ. Draft editing and review were done by all authors.

*Competing interests.*

The contact author has declared that neither they nor their co-authors have any competing interests.

*Disclaimer.*

*Acknowledgements.*

The authors would like to thank the editor-in-chief, associate editor, and the two anonymous reviewers for their effort and help during the review process.

*Financial support.*

David Chaparro was supported by 'Fundació La Caixa' project LCF/BQ/PI23/11970013, and by the H2020 Project FORGENIUS (Improving access to FORest GENetic resources Information and services for end-USers). María Piles thanks the support of *Conselleria de Innovación, Universidades, Ciencia y Sociedad Digital* through the project AI4CS CIPROM/2021/56.

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
