# Peer review of "Assessing evapotranspiration dynamics across central Europe in the"

_EGUsphere, 2024_

## Author Comment (AC1)

Response to Referee #1

**Assessing evapotranspiration dynamics across central Europe in the context of land-atmosphere drivers**

| Comments from Reviewer 1 | |
|---|---|
| The manuscript titled **"Assessing evapotranspiration dynamics across central Europe in the context of land-atmosphere drivers"** evaluates evapotranspiration (ET) products derived from satellite remote sensing, modeling, and reanalysis data in conjunction with in-situ observations from Integrated Carbon Observation System (ICOS) stations in central Europe from 2017 to 2020. The study investigates the effects of varying land cover types, soil moisture (SM), and vapor pressure deficit (VPD) on ET dynamics, including the severe drought of 2018. It uses extended triple collocation methods to assess the accuracy of ET products, revealing notable differences among products under heterogeneous land cover conditions and during drought years. The findings highlight that ET variability is strongly influenced by VPD under non-limiting soil moisture conditions and demonstrate that SEVIRI, ERA5-land, and GLEAM products show superior performance. The research provides insights into the suitability of various ET products for capturing land-atmosphere interactions and drought impacts across diverse land cover types. This manuscript presents valuable insights into evapotranspiration dynamics across central Europe, but significant revisions are needed before it can be considered for publication. | Thank you very much for all efforts and help to improve the manuscript. We tried our best to answer all comments satisfactorily. |
| **Major comments:**

   1. Limitations in the Performance Analysis of ET Products | We discuss performance differences among ET products regarding varying climatic conditions, landcover types, and retrieval methods throughout the manuscript and especially in sec. 4. However, we can certainly add more information about the practical implications and objective of this study in sec. 4. and 5. of the manuscript. |

| | |
|---|---|
| The manuscript reveals significant performance differences among ET products, but it does not fully address how these discrepancies might affect the conclusions drawn, particularly under specific land cover types or climatic conditions. A deeper analysis of how these differences impact the study's practical implications is crucial. | In order to provide additional deeper analyses as suggested by the reviewer, we want to suggest two potential analyses, which we could conduct:

1. Provide statistics (e.g., variance) for figures 10 & 11 of the manuscript, in order to analyze in more detail the impact of the drought year.
2. Provide power spectrum analyses for different time frames for every ET product (e.g., daily, bi-weekly, monthly) to analyze performance and biases of every product along time. |
| 3. In-depth Analysis of Drought Year Effects

The analysis of the 2018 drought is valuable, but the study does not sufficiently explore how different ET products capture the effects of drought. A more detailed comparison of how products perform under drought conditions is needed, including which models and parameterizations are more sensitive to such extremes. | We discussed the impacts of the drought year on ET products in sec. 4.2., e.g., lines 518-526 or lines 538-546. However, related to the comment above, we suggest to add statistics to the binning plots (Figs. 10 & 11) in order to provide deeper analyses of drought year effects. |
| 1. Improvement of Remote Sensing and Ground Observation Matching

There is an acknowledged mismatch between point-scale ICOS data and the coarser-resolution remote sensing products. The manuscript could benefit from a discussion on methods to address this issue, such as spatial downscaling or data fusion techniques, to improve the alignment of ground and remote sensing observations. | In the field of remote sensing, the spatial mismatch between satellite data and ground measurements is a well-known problem and somehow accepted assumption in order to be able to validate remote sensing products. In that sense, we agree on providing some discussion.

We will add an explanation why we kept coarse-scale ET products at their original spatial resolution after line 174:

'The MODIS product with nominal spatial resolution of 500 m is aggregated to the 3 km footprint, while the SEVIRI, ERA5-land, GLDAS-2, and GLEAM products are maintained at their original spatial resolutions of 3 km, 9 km and 25 km, respectively. **Although there exist several downscaling methods and data fusion techniques for improving the spatial resolution of remote sensing products (Ha et al., 2013; Mahour et al., 2017; Peng et al., 2017), we decided to keep ET products with a spatial resolution lower than 3 km at their original resolution (i.e., GLDAS-2 and GLEAM at 25 km), since, for one, the intention of this study is a comparison of well-known and established ET products and not an optimization of rescaled comparisons. Second, because we did not want to include additional uncertainties which potentially originate from the employed downscaling method or auxiliary datasets. Every downscaling approach has its advantages and drawbacks as it intends to statistically correlate coarse-scale data and fine-scale auxiliaries or develop physically-based models to enhance the spatial resolution (Peng et al., 2017).**' |

| | |
|---|---|
| | Further, we will add a discussion addressing this comment after line 470: |
| | 'One reason for the mismatch between the ICOS product and SEVIRI, GLEAM and ERA5-land is surely the spatial mismatch between the point-scale ground-based EC tower measurements and the remote sensing (3 km) or reanalysis (9 km) products. To capture vegetation stress, ecosystem health, and fine-scale variability in ET globally, adequate spatial (and temporal) resolutions are necessary. **Here, detailed research regarding downscaling techniques as reviewed in, e.g. Mahour et al., 2017; Peng et al., 2017, that combine medium-scale ET data with fine-scale auxiliaries in order to improve the spatial resolution are needed regarding its uncertainties and impact on product comparisons**.' |
| | Lastly, we draw a conclusion regarding this topic after line 577: |
| | 'In summary, when considering all conducted analyses together (spatial and temporal resolutions, product dependencies, ETC results, SM and VPD controls on ET), the remote sensing products SEVIRI and GLEAM as well as reanalysis product ERA5-land seems to provide most reasonable results compared to all other ET products, with SEVIRI providing a higher temporal and spatial resolution compared to GLEAM and ERA5-land. **Hence, despite their coarse spatial resolution, GLEAM and ERA5-land are able to capture ET dynamics sufficiently even under drought conditions. Future research regarding data fusion techniques and downscaling approaches that combine coarse- or medium-scale ET data with fine-scale auxiliaries in order to improve the spatial resolution of certain ET products may help to decrease the spatial mismatch and optimize the comparison between point-scale field measurements and satellite remote sensing or modelling data.**' |
| | Even if we could largely extend the discussion on these methods, we prefer to limit it to the paragraphs written above since research regarding improved matching of remote sensing observations with ground measurements is out of the scope of this manuscript. |
| 1. Considering extending the study period

The study period appears to be too short, and the comparison between satellite-based ET and local measurements seems insufficient. The authors might consider extending the study period and incorporating a comparison across different timeframes. This would enable the use of the | We decided on purpose to analyze the years 2017 to 2020, for one, due to the availability of all datasets, and second, because we wanted to include detailed analyses regarding multiple ET products and varying climatic, landcover and site conditions. Further, we do not think that adding more years is necessary to draw conclusions on how well different ET products capture ET dynamics and drought events, as we already included varying climatic (one drought (2018) and one wet year (2017)) as well as landcover and site conditions (eight stations with different landcover and climate conditions as well as levels of landcover heterogeneity).

Analyzes across different time frames and evaluation regarding the water balance principle is out of the scope of the manuscript, which |

| | |
|---|---|
| water balance principle for a more comprehensive evaluation. | intends to compare well-known and established ET products that are commonly used in climatic research studies. |
| **Minor comments**

1. At lines 34-35: please add explanation (why). | The explanation is found in the following sentence. We can rephrase the sentences to provide the explanation right away:

'The greatest deviations are found at the agricultural-managed sites Selhausen (Germany) and Bilos (France), with the former also showing the highest potential dependencies (error cross-correlation) between the ET products. **Hence, o**ur results indicate that ET products differ most at stations with spatio-temporal varying land cover conditions (varying crops over growing periods and between seasons).' |
| 2. At line 34: please add the specific number to describe "the highest potential dependencies (error cross-correlation)". | Of course, we can add the specific number:

'The greatest deviations are found at the agricultural-managed sites Selhausen (Germany) and Bilos (France), with the former also showing the highest potential dependencies (error cross-correlation **(ECC)**) between the ET products **(up to 7.6 and outside the acceptable range of -0.5 < ECC < 0.5)**.' |
| 3. At lines 36-38, please explain why. | The explanation is found in the following sentence.
Due to the more complex heterogeneity of land covers at agricultural stations, the estimation of ET is more complex and challenging, while needle-leaved stations have less complexity due to rather stable vegetation conditions. We can rephrase the sentences to be more precise:

'This **is since** complex heterogeneity complicates the estimation of ET, while ET products agree **well** at evergreen needle-leaved stations with less temporal changes throughout the year and between years.' |
| 4. Please add the longitude and latitude in Figure 1. | Done. We will add coordinates to the map in figure 1. |
| 5. Please add a table that describes all the sites, including their longitude, latitude, land cover type, altitude, and other relevant details. | Done. We will add a table S1 in the supplement providing the coordinates, ecosystem type, altitude, climate zone and mean annual precipitation and air temperature for all stations. Details on percentages of different land cover types for every station are already given in supplement table S2 (formerly tab. S1). |
| 6. At line 150: Please explain why you used the standardized precipitation-evapotranspiration index (SPEI). | Done. We can add some sentences to explain why we used the SPEI instead of SPI or other indices after line 153:

'**We choose to use the SPEI to identify drought conditions instead of the standardized precipitation index (SPI) or other indices (i.e., Palmer drought severity index), since the SPEI additionally considers temporal changes in ET and hence, temperature, which showed to be evident for identifying abnormal (drought) conditions. Previous studies showed that not only the lack of precipitation defines drought events but also the level of temperature and consumption of rainfall by evaporation and/or transpiration (Vicente-Serrano et al., 2010).**' |

| | | |
|---|---|---|
| 7. | If the time step is from hourly to 8-day, consider generated them at a shorter time step rather than monthly which loose too much information. | We agree that analyzing monthly time steps would be temporally too coarse. That is why in the manuscript, we are analyzing daily and 8-daily time steps, never monthly. |
| 8. | At lines 206-225: The spatial resolution differences among these ET products are quite significant. The authors should consider using methods to standardize all the ET products to a common resolution or use products with longer time periods. Otherwise, direct comparisons may not be valid or meaningful. | This comment is related to comments #3 and #4 above, where we provide our changes implemented in the manuscript to account for them.

We agree that the change across spatial resolutions between different products are quite significant and we tried to account for it to certain degree as we upscaled high-scale ET products (e.g., MODIS at 500 m) to the baseline resolution of 3 km. And we decided to keep coarse-scale ET products (> 3 km, e.g., GLDAS-2 and GLEAM at 25 km) at their original resolutions instead of rescaling, since, for one, the downscaling from 25 km to 3 km would be a very big step and we did not want to include additional uncertainties from the employed rescaling method, and second, end users who would use only one ET product within their analyses alongside other variables would not rescale the ET product to one specific resolution and hence, loose the conclusions drawn from our analyses if we would compare all products at one rescaled resolution.

We think the comparison is meaningful and valid as we compare well-known and employed ET products under varying climatic conditions and for multiple landcover classes at several stations all across Central Europe. And we were able to show that despite the coarse spatial resolution of GLEAM, the product is able to capture ET dynamics quite well compared to other products, which is an important conclusion for the scientific community and especially for studies that are based on the GLEAM ET product only. |
| 9. | Table 1: please change as three lines table format and added other dataset's information such as soil moisture and SPET etc. | Done, we included other products (VPD, SM, SPEI) in table 1. However, we did not change the format of the table as we like it this way. |
| 10. | At lines: 228-233 The removal of the seasonal signal may not be necessary, as the study period is too short for such adjustments. | We understand this point and agree that for longer time series, detrending has much more impact. We would like to keep the anomalies in order to compare detrended ET time series, e.g., figure 8 & 9, even if the time series considered is rather short compared to real climatological analyses. |
| 11. | At lines 295-296: Please provide the specific values for the highest $R^2$ and lowest RMSE and lowest percentage bias, PBIAS here. | We provide the detailed numbers for $R^2$, RMSE and PBIAS in the following sentences (lines 297-300). Since every station provides multiple statistics between all products (as shown in supplement figures S5-S7), we first wanted to highlight specific stations with outstanding statistics and then provide more details with actual numbers for the products. Otherwise, the sentences would get very long and complex, hence, we thought it might be easier for the reader to follow. |

| | |
|---|---|
| 12. Figure 4. Please added some statices numbers in every panel. | Figure 4 shows the time series of all ET products and for all stations. Meaning, we are showing one single time series of every ET product and for every station individually. We are not sure what kind of statistics would make sense for single time series that would provide any useful information. |
| 13. Same as Figure 4 | We guess this comment is related to figure 5 and that we should add statistics in figure 5, same as in figure 4? Or to which figures is this comment related to?

The statistics (correlation, error & percentage bias) for figure 5 as well as similar supplement figures S2-S4 are given in supplement figures S5-S7. Since the figures give already a lot of information, we provide the statistics in separate figures. |

---

## Author Comment (AC2)

Response to Referee #2

**Assessing evapotranspiration dynamics across central Europe in the context of land-atmosphere drivers**

| Comments from Reviewer 2 | |
|---|---|
| Review Report for Manuscript ID: egusphere-2024-3386

Title: Assessing Evapotranspiration Dynamics Across Central Europe in the Context of Land-Atmosphere Drivers

General Comments

This study provides a comprehensive evaluation of evapotranspiration (ET) products across central Europe using a combination of in-situ, remote sensing, and reanalysis datasets. The authors analyzed the performance of multiple ET datasets under different climatic conditions, particularly focusing on the severe drought. The study effectively addresses a research gap by assessing the agreement and discrepancies between ET products in the context of soil moisture (SM) and vapor pressure deficit (VPD) interactions. The manuscript is relevant for researchers studying land-atmosphere interactions, hydrology, and ecosystem responses to climate extremes. However, several key areas require further clarification and improvement to strengthen the manuscript before publication. The authors should clarify the rationale for dataset selection, improve the discussion on physical interpretability, and provide additional insights into the role of vegetation stress and uncertainty quantification. | Thank you very much for reviewing our study and for outlining the relevance of the manuscript. We tried our best to provide satisfactorily answers to all comments in order to improve and strengthen the manuscript. |
| Major Comments

Justification for Selected ET Products, the authors compare ET estimates from various remote sensing and modeling products (MODIS, SEVIRI, GLEAM, ERA5-land, GLDAS). However, it would be beneficial to explicitly justify the selection of these specific products over other alternatives such as FLUXCOM, ETMonitor, EB-ET. MODIS is kind of ET more relying | Thank you for naming several other ET products and providing the reference for the EB-ET product.

The reviewer mentioned products are higher order ET products derivated from either other ET products or satellite products (e.g., Fluxnet ET observations, MODIS), and often provide data outside the time period analyzed in this study. In this study, we compare ET products directly retrieved from the observations and mostly from one single sensor (e.g., ICOS LE observations, MODIS optical data, |

| | |
|---|---|
| on optical data. GLEAM is based on microwave data. Please check other thermal ET product. One could be EB ET. Chen et al. 2021, Remote sensing of global daily evapotranspiration based on a surface energy balance method and reanalysis data. Journal of Geophysical Research: Atmospheres, 126(16): e2020JD032873. | ERA5-land reanalysis, SEVIRI) between 2017-2020. However, we can add the proposed products to our list, when we introduce existing ET products, and further discuss their advantages and disadvantages. |
| | We will add sentences to justify in more detail why we choose these ET products (most commonly known and employed ones) and acknowledge the existence of other products, e.g.,: |
| | Line 104: 'In this study, we first compare the most common ET products from field measurements, modelling, and remote sensing across 104 central Europe for the period 2017 to 2020. **These selected products are well-known, commonly employed, and freely available.**' |
| | Line 488: 'Although there exist other ET products from remote sensing and modelling, e.g., (Jiménez et al., 2011; Mueller et al., 2013; Fisher et al., 2020; De Santis et al., 2022; Yu et al., 2023), the examined ET products in this study are appropriate when addressing global analyses since other products have either a more coarse spatial or temporal resolution (Yu et al., 2023), are limited to clear sky conditions (De Santis et al., 2022), which prohibits continuous time series of ET measurements, **or are higher order derivates from either field measured or merged remote sensing based products (Jung et al., 2019; Chen et al., 2021).**' |
| Additionally, the manuscript should discuss the potential biases associated with the retrieval algorithms used in each dataset and how these may affect ET estimates under different climatic conditions. | Certainly, we will add a paragraph for discussing on potential biases associated with the retrieval algorithms for every ET product. |
| Physical Interpretability and Model Dependencies, the study provides robust statistical comparisons but lacks a deeper discussion on the physical implications of the observed differences. For example, why do some products perform better at evergreen needle-leaved sites compared to agricultural sites? How do land cover heterogeneity and seasonal changes influence model uncertainties? | We discuss the performance of the different ET products for varying landcover classes throughout the manuscript, and discuss the influence of land cover heterogeneity on retrieval results in section 4. of the manuscript. More thorough analyses and discussion regarding the physical interpretability and model dependencies with that many ET products would lengthen the paper significantly, which is impractical and potentially more interesting when focusing on two or three ET products. |
| Since GLEAM incorporates reanalysis and satellite-based observations, its correlation with other datasets like ERA5-land and GLDAS-2 might be inflated. Have the authors accounted for interdependencies between products in their error analysis? | We accounted for the interdependencies between products and calculated the error cross-correlation (ECC) between all products (see sec. 2.3.1.) in order to statistically validate the interdependencies. We discuss this in sec. 3.1. as well as 4.1., and give the ECC results in supplement figure S8. |

| | |
|---|---|
| The role of vegetation stress and physiological controls (e.g., stomatal closure) in driving ET reductions during drought should be better discussed, perhaps using additional to support this point. | Certainly, we will add more discussion on the role of vegetation stress and physiological controls in ET reductions during the drought year 2018. |
| Evaluation of Uncertainty and Error Cross-Correlation (ECC), The extended triple collocation (ETC) analysis is a valuable approach, but some ECC values are quite high, particularly at agricultural sites. The manuscript should explicitly discuss how ECC influences the reliability of the results and whether certain datasets may be inherently dependent. | We discussed the ECC results and potential interdependencies between ET products in sec. 3.1. as well as 4.1., and showed that ETC results are uncorrelated to ECC results, since at station DE-Rus, which gave high ECC results (potential strong interdependencies), ETC results do not reflect any interdependencies. |
| Clarity of Figures and Statistical Significance, the scatter plots and time series comparisons are informative, but additional clarity is needed in figures showing product inter-comparisons (e.g., Figures 4, 5, 6). Including a statistical significance test for differences between ET products would enhance the rigor of the results. | We included significance tests in figures 8 and 9 since we agree that some analyses need statistical significance information to discuss the results reliably.

However, figure 4 shows the time series of all ET products and for all stations. Meaning, we are showing one single time series of every ET product and for every station individually. We are not sure what kind of statistics would make sense for single time series that would provide any useful information.

In figure 5, we show the scatterplots, for which we give the Pearson's correlation coefficient ($R^2$), root-mean square error (RMSE) and percentage bias (PBIAS) between each product and for each station in supplement figures S5-S7, which we discuss in sec. 3. and 4. |
| Minor Comments

Grammar and Style: Some sentences are long and complex, making them difficult to follow. Consider simplifying and improving readability. For example: The ICOS network has undertaken a large effort to ensure high-quality LE measurements, which are comparable among different ICOS stations.". Suggested revision: "The ICOS network has made significant efforts to ensure consistent high-quality LE measurements across stations.". Line 555 Grammar mistake, This is, products were most consistent with each other at stations with less complex land cover conditions and changes throughout the seasons (the evergreen needle-leaved stations DE-Ruw and FI-Let). | Well taken, we can certainly improve the grammar and style. |
| Line 49, rephrase the sentence 'Since precipitation (P) 'and evaporation are the two key components of the global water cycle' (Miralles et al., 2011), another important proxy for analyzing water stress and | Done, we can rephrase the sentence:

'Evapotranspiration (ET) is an important proxy for analyzing water stress and its effects on ecosystems since |

| | |
|---|---|
| its effects on ecosystems is evapotranspiration (ET).' | precipitation (P) 'and evaporation are the two key components of the global water cycle' (Miralles et al., 2011).' |
| Line80, optical, thermal, infrared, or microwave observations are used to derive ET based on surface energy balance, physical and empirical models (Bayat et al., 2021, 2024; Rahmati et al., 2020; Zhang et al., 2016). The cited reference does not include thermal observation based ET from surface energy balance method. | Thank you for pointing this out. We will include two additional references, explicitly addressing thermal based ET and the surface energy balance method: |
| | 'Although it is not directly measurable from remote sensing acquisitions, optical, thermal, infrared, or microwave observations are used to derive ET based on surface energy balance, physical and empirical models (Zhang et al., 2016; Rahmati et al., 2020; **Singh et al., 2020;** Bayat et al., 2021; **Bhattacharya et al., 2022;** Bayat et al., 2024).' |
| | These references are: |
| | Singh, R.P., Paramanik, S., Bhattacharya, B.K. *et al.* Modelling of evapotranspiration using land surface energy balance and thermal infrared remote sensing. *Trop Ecol* **61**, 42–50 (2020). https://doi.org/10.1007/s42965-020-00076-8 |
| | Bhattacharya, B. K., Mallick, K., Desai, D., Bhat, G. S., Morrison, R., Clevery, J. R., Woodgate, W., Beringer, J., Cawse-Nicholson, K., Ma, S., Verfaillie, J., and Baldocchi, D.: A coupled ground heat flux–surface energy balance model of evaporation using thermal remote sensing observations, Biogeosciences, 19, 5521–5551, https://doi.org/10.5194/bg-19-5521-2022, 2022. |
| Terminology Consistency: The terms "ET estimation," "ET retrieval," and "ET modeling" are used interchangeably. It would be beneficial to define them more precisely and use consistent terminology throughout the manuscript. | Done. We will use one terminology (ET estimation) throughout the manuscript. The term 'modelling' is only used for modelled ET products (e.g., GLDAS-2). The term 'retrieval' is only used when talking about the retrieval algorithm/method of the ET products. |
| Temporal Aggregation Effects: Some ET product has a lower temporal resolution than other datasets. Have the authors checked whether this affects the observed discrepancies, if upscaled to 15 days, even monthly temporal resolution? | Yes, we have checked the effect of temporal aggregation. In the manuscript, we included some of these analyses in figure 9 and supplement figure S10 by comparing the Kernel density estimates of ET anomalies on daily and 8-daily time scales. Here, we clearly see a difference in the density curves and hence, included it in the publication as these are the main temporal scales included in the paper. As most products provide high temporal resolution (< daily) except for MODIS, which provides 8-daily ET data, analyzing other temporal scales (15-daily, monthly) is out of the scope of the current manuscript. |
| Line 558, The authors wrote that: The remote sensing products, SEVIRI, MODIS, and GLEAM, performed equivalently well or even better than the in-situ | Thank you for pointing this out. We are not saying that remote sensing products are better than in-situ observations. What we mean is, that for our specific study design |

| measured (ICOS), I don`t understand why the remote sensing products can be better than in-situ measured data? This is confusing readers. How can a satellite ET product be better than measurement? | (3 km footprint, daily analyses), the performance of the remote sensing products is overall more comparable and consistent among all investigated datasets, including ICOS. We discussed the reasons for this, e.g., in lines 465-478, also providing the drawbacks of in-situ measured ET observations at ICOS stations. We can rephrase the sentence in line 558 in order to clarify the meaning. |
|---|---|
| | 'The remote sensing products, SEVIRI, MODIS, and GLEAM, performed equivalently well or even better than the in-situ measured (ICOS), modelled (GLDAS-2) or reanalysis (ERA5-land) products **for this specific study concept (3 km footprint, daily analyses)**.' |
| Line 514, ET is more controlled by atmospheric demand rather than atmospheric supply, I can understand when the atmosphere is warming, it will need more vapor evaporated from ground. This could be a kind of atmospheric demand, but do not understand what is atmospheric supply? What kind of supply from atmosphere can influence ET? Are you saying precipitation? Please rephrase this sentence to make it clear. | Thank you for pointing this out. We actually meant soil water supply (soil moisture) as we see no dependency of ET on SM but variations of ET with VPD during wet years. We can rephrase the sentence accordingly: |
| | 'Here, our results indicate that ET is more controlled by atmospheric demand rather than water supply from atmosphere **(precipitation) and soil (soil moisture)** as reported also by Zhou et al., (2019).' |
| Line 523, Further, results show that VPD and SM are negatively coupled during extreme events as reported also by (Zhou et al., 2019)-à by Zhou et al. 2019. Same as reported by (De Santis et al., 2022). | Thank you, we will include De Santis et al., 2022 as additional reference. |
| Section 4, there are many other global ET product, which are not discussed. Please check and compare them. | As mentioned in our second answer above, we will mention other ET products and include additional information why we choose to compare these ET products. |

---

## Referee Report (RR1)

**Recommendation: Minor Revision**

The revised manuscript shows clear improvement in both structure and scientific clarity compared to the original submission. The authors have responded to most of the previous concerns appropriately, and the current version is more coherent, with enhanced data presentation and interpretation. However, several minor issues remain that should be addressed before final acceptance. My detailed comments are as follows:

1. **Overall evaluation**

   The newly revised manuscript demonstrates substantial improvement over the previous version, particularly in terms of organization, clarity of data presentation, and interpretation of results. The current version is more scientifically rigorous and readable.

2. **Line 26**

   Please provide the data sources for the two datasets mentioned. Citing the origin of the data is essential for reproducibility and transparency.

3. **Lines 122 and 145**

   "table S1" should be capitalized as "Table S1" in both instances to conform to academic writing standards.

4. **Lines 281–283**

   This section requires a more specific and nuanced explanation. The current sentence vaguely attributes the underestimation and delayed ET rise to spatial differences and vegetation dependence. I suggest the authors clarify that **remote sensing data loss during frequent precipitation events** could be a contributing factor to the underestimation and lag in ET rise. For instance, at **DE-Hai (broad-leaved forest)** and **DE-Ruw (coniferous forest)**, ICOS ET remains consistently lower than other products and shows a delayed seasonal increase. These discrepancies may be due to **differences in the spatial resolution of the ET products and their sensitivity to vegetation phenology**.

5. **Lines 487–488**

The statement "Reason for that are… reduced transpiration of agricultural sites throughout the year compared to forested sites" is too general and potentially oversimplified. Please provide a more detailed explanation. For example:

*"DE-Rus, classified as an agricultural site located in a non-irrigated zone, shows relatively low vegetation cover (e.g., mean NDVI value if available). This can lead to underestimation of ET in that pixel when using models that rely on vegetation indices. Combined with the site's seasonal vegetation dynamics and lack of irrigation, this explains the lower ET values compared to forested areas with more consistent canopy cover."*

---

## Author Response (AR2)

Response to Referee #1

**Assessing evapotranspiration dynamics across central Europe in the context of land-atmosphere drivers**

| Comments from Reviewer 1 | |
|---|---|
| **Recommendation: Minor Revision**

The revised manuscript shows clear improvement in both structure and scientific clarity compared to the original submission. The authors have responded to most of the previous concerns appropriately, and the current version is more coherent, with enhanced data presentation and interpretation. However, several minor issues remain that should be addressed before final acceptance. My detailed comments are as follows: | Thank you very much for your positive feedback. We gladly addressed the remaining issues. |
| 1. Overall evaluation: The newly revised manuscript demonstrates substantial improvement over the previous version, particularly in terms of organization, clarity of data presentation, and interpretation of results. The current version is more scientifically rigorous and readable. | Thank you very much. We are glad to hear that our manuscript improved to your satisfaction. |
| 2. Line 26: Please provide the data sources for the two datasets mentioned. Citing the origin of the data is essential for reproducibility and transparency. | Done. We included the data sources for soil moisture and water vapor pressure deficit. |
| 3. Lines 122 and 145: "table S1" should be capitalized as "Table S1" in both instances to conform to academic writing standards | Done. We checked again all references to tables and figures and wrote them all with capital letters. |

| | |
|---|---|
| 4. Lines 281–283: This section requires a more specific and nuanced explanation. The current sentence vaguely attributes the underestimation and delayed ET rise to spatial differences and vegetation dependence. I suggest the authors clarify that **remote sensing data loss during frequent precipitation events** could be a contributing factor to the underestimation and lag in ET rise. For instance, at **DE-Hai (broad-leaved forest)** and **DE-Ruw (coniferous forest)**, ICOS ET remains consistently lower than other products and shows a delayed seasonal increase. These discrepancies may be due to **differences in the spatial resolution of the ET products and their sensitivity to vegetation phenology**. | We added a text paragraph (280-284) to explain this in more detail: 'These differences and delayed seasonal increase of remote sensing, modelling, and reanalysis products compared to the ICOS measurements at the DBF and ENF station occur, for one, due to the discrepancies in spatial resolutions (point-scale versus kilometer scale). Second, ICOS field measurements provide a different sensitivity to vegetation phenology than the other remote sensing & modelling products due to measuring directly above the canopy.' |
| 5. Lines 487–488: The statement "Reason for that are… reduced transpiration of agricultural sites throughout the year compared to forested sites" is too general and potentially oversimplified. Please provide a more detailed explanation. For example: *"DE-Rus, classified as an agricultural site located in a non-irrigated zone, shows relatively low vegetation cover (e.g., mean NDVI value if available). This can lead to underestimation of ET in that pixel when using models that rely on vegetation indices. Combined with the site's seasonal vegetation dynamics and lack of irrigation, this explains the lower ET values compared to forested areas with more consistent canopy cover."* | Thank you for the reviewer's advice. We added a more detailed explanation to this statement in lines 489-493, following your example: 'Further, the mostly non-irrigated arable land at station DE-Rus (see Fig. 2) shows relatively low vegetation cover (LAI < 2; normalized difference vegetation index (NDVI) around 0.5 during summer months (not shown)) compared to forested sites (LAI > 5, NDVI around 0.8 during summer months (not shown)), which can lead to an underestimation of ET when using models that rely on vegetation indices (i.e., NDVI, LAI). Combined with the seasonal vegetation dynamics of this station and the lack of irrigation, this explains the lower ET values compared to forested areas with more consistent canopy cover.' |